



# Design and Analysis of a Wake Steering Controller with Wind Direction Variability

Eric Simley[1], Paul Fleming[1], and Jennifer King[1]

[1]National Wind Technology Center, National Renewable Energy Laboratory, Golden, CO, 80401, USA

*Correspondence to:* Eric Simley (eric.simley@nrel.gov)

**Abstract.** Wind farm control strategies are being developed to mitigate wake losses in wind farms, increasing energy production. Wake steering is a type of wind farm control in which a wind turbine's yaw position is misaligned from the wind direction, causing its wake to deflect away from downstream turbines. Current modeling tools used to optimize and estimate energy gains from wake steering are designed to represent wakes for fixed wind directions. However, wake steering controllers must operate
in dynamic wind conditions and a turbine's yaw position cannot perfectly track changing wind directions. Research has been conducted on robust wake steering control optimized for variable wind directions. In this paper, the design and analysis of a wake steering controller with wind direction variability is presented for a two-turbine array using the FLOw Redirection and Induction in Steady State (FLORIS) control-oriented wake model. First, the authors propose a method for modeling the turbulent and low-frequency components of the wind direction, where the slowly varying wind direction serves as the relevant input
to the wake model. Next, we explain a procedure for finding optimal yaw offsets for dynamic wind conditions considering both wind direction and yaw position uncertainty. We then performed simulations with the optimal yaw offsets applied using a realistic yaw offset controller in conjunction with a baseline yaw controller, showing good agreement with the predicted energy gain using the probabilistic model. Using the Gaussian wake model in FLORIS as an example, we compared the performance of yaw offset controllers optimized for static and dynamic wind conditions for different turbine spacings and turbulence in-
tensity values. For a spacing of 5 rotor diameters and a turbulence intensity of 10%, robust yaw offsets optimized for variable wind directions yielded an energy gain improvement of 128%. In general, accounting for wind direction variability in the yaw offset optimization process was found to improve energy production more as the separation distance increased, whereas the relative improvement remained roughly the same for the range of turbulence intensity values considered.

## Copyright Statement

The author's copyright for this publication is transferred to Alliance for Sustainable Energy, LLC. This work was authored by the National Renewable Energy Laboratory, operated by Alliance for Sustainable Energy, LLC, for the . for the U.S. Department of Energy (DOE) under Contract No. DE-AC36-08GO28308. Funding provided by the U.S. Department of Energy Office of Energy Efficiency and Renewable Energy Wind Energy Technologies Office. The views expressed in the article do not necessarily represent the views of the DOE or the U.S. Government. The U.S. Government retains and the publisher, by
accepting the article for publication, acknowledges that the U.S. Government retains a nonexclusive, paid-up, irrevocable,



worldwide license to publish or reproduce the published form of this work, or allow others to do so, for U.S. Government purposes.

# 1 Introduction

Wind farm control is a type of control strategy in which wind turbines are actuated to influence the aerodynamic interaction
between turbines in a wind farm, thereby improving the total energy production or reducing structural loads (Johnson and Thomas (2009); Boersma et al. (2017)). Although several methods of actuation exist for influencing the wake behind a wind turbine (Fleming et al. (2014); Boersma et al. (2017)), one of the most effective and easily implementable strategies for increasing energy production being explored is wake steering (Dahlberg and Medici (2003); Wagenaar et al. (2012)). Wake steering control involves intentionally misaligning turbines' nacelle positions relative to the wind direction, thereby steering
their wakes away from downstream wind turbines. Although the misaligned turbines generate less power, the total power produced by the wind farm can be increased as a result of the higher wind speeds experienced by downstream turbines.

Wake steering control has been studied using computational fluid dynamics (CFD), wind tunnel experiments, and full-scale field experiments. Wake steering was shown to increase the total power production of a six-turbine wind farm using large-eddy simulation (LES), a type of CFD, by Gebraad et al. (2016). Additionally, Vollmer et al. (2016) used LES to investigate the
impact of different atmospheric stability conditions on the effectiveness of wake steering. Using two-turbine arrays comprised of scaled wind turbines in a wind tunnel, Campagnolo et al. (2016) and Schottler et al. (2016) also demonstrated an overall increase in power production with wake steering. Recently, wake steering experiments at commercial wind farms have suggested that an increase in total energy production is realizable in the field for a two-turbine scenario (Fleming et al. (2017, 2019)), with Fleming et al. (2019) observing an average energy increase of 4% for a turbine pair over wind directions where wake
steering is active. Although high-fidelity modeling and experiments are necessary to validate wake steering, computationally efficient engineering models of wake steering are needed for optimizing controllers and estimating wind farm energy production. For example, the FLOw Redirection and Induction in Steady State (FLORIS) tool developed by the National Renewable Energy Laboratory (NREL) and Delft University of Technology (Gebraad et al. (2016)) provides a framework for optimizing wake steering strategies, allowing the user to choose between several different engineering wake models. The FLORIS code is
available at https://github.com/NREL/floris (NREL (2019)) with documentation provided at https://floris.readthedocs.io.

Analyses of wake steering using CFD simulations, wind tunnel tests, and engineering models such as FLORIS are useful for demonstrating the effectiveness of wake steering, but are typically performed assuming fixed wind directions and yaw positions. In reality, large-scale weather phenomena cause the mean wind direction across the wind farm to vary over time. Wind turbines are unable to perfectly track the changing wind directions because of typically slow yaw controller dynamics
as well as difficulty estimating the wind direction from noisy measurements. This is even more important when implementing wind farm control, wherein the wind direction must be estimated from imperfect measurements by a wake steering controller to determine the appropriate yaw offset to apply. Because of wind direction variability, slow yaw controller dynamics, and the uncertainty inherent in yaw control, energy gains from wake steering are expected to be lower in the field than predicted by





analyses assuming static wind directions and yaw positions. To address wind direction variability, Bossanyi (2018) performed wind farm control simulations using a dynamic simulation model with time-varying wind conditions, highlighting controller design choices relevant to dynamic wind conditions. However, the applied yaw offsets are optimized assuming static wind directions. Wind direction variability is analyzed in a statistical sense by Gaumond et al. (2014), who show that using a wake

model to accurately predict wake losses in a wind farm for a specific mean wind direction requires wind direction variability about the mean direction to be considered. To optimize a wake steering strategy for energy production, Quick et al. (2017) use optimization under uncertainty (OUU) to find yaw offset targets that maximize energy production when there is uncertainty in the achieved yaw position. Rott et al. (2018) similarly use an OUU approach to optimize yaw offsets for energy production considering variability and uncertainty in the wind direction during periods of constant yaw position. Using the FLORIS wake

model, both Quick et al. (2017) and Rott et al. (2018) show that robust wake steering strategies accounting for yaw or wind direction uncertainty typically involve lower-magnitude yaw offsets yet outperform "static-optimal" wake steering strategies, which are optimized for fixed wind directions, when uncertainty exists.

This article builds on the work of Quick et al. (2017) and Rott et al. (2018) by including both wind direction uncertainty, resulting from wind direction variability, and yaw position uncertainty in the robust yaw offset optimization process. An

additional contribution of this work is to quantify wind direction and yaw position uncertainty using realistic yaw and yaw offset control simulations with stochastic wind direction signals based on field measurements. However, rather than directly using a turbulent wind direction signal to determine wind direction and yaw position uncertainty, the authors propose a method for deriving a slowly varying wind direction time series representing the time-varying mean wind direction across the wind farm without turbulence. This low-frequency wind direction signal acts as a more relevant input to the FLORIS wake model,

which already contains the effects of turbulence for a fixed mean wind direction. The developed method for optimizing yaw offsets with wind direction and yaw position uncertainty is demonstrated using the example of a two-turbine array (a scenario of interest for initial field validation studies) with the Gaussian wake model in FLORIS (Annoni et al. (2018b)). An additional contribution of this research is to evaluate the energy gains achieved by the robust "dynamic-optimal" yaw offsets using realistic wake steering control simulations, which show close agreement with the energy gains predicted from the probabilistic

model of wind direction and yaw uncertainty. Finally, by varying the turbine spacing and turbulence intensity, where the latter affects the degree of wake expansion/recovery in the Gaussian wake model, wind direction variability is shown to become more important in the yaw offset optimization process as the separation distance increases, but has roughly the same impact for different turbulence intensity values.

The rest of the article is organized as follows. Section 2 describes the models used in the research, including the wake,

wind turbine, yaw controller, wake steering controller, and wind direction models, as well as the wake steering simulation procedure. The procedure for quantifying wind direction and yaw uncertainty as well as optimizing yaw offsets for the case of uncertain wind directions and yaw positions is described in Section 3. Using the Gaussian wake model in FLORIS, Section 4 contains the results of wake steering controller simulations for a two-turbine array in dynamic wind conditions for both static and dynamic-optimal yaw offsets, highlighting the improvement in energy gain when yaw offsets are optimized considering

wind direction and yaw uncertainty. Sections 4.2 and 4.3 show the dependence of wake steering with wind direction variability



on turbine spacing and turbulence intensity. Last, further discussion of the results is provided in Section 5, which concludes the paper.

## 2 Models

This section provides a description of the wake model, wind turbine model, the yaw and yaw offset controllers, as well as
the dynamic wind direction model and the simulation procedure used in the analysis of wake steering with wind direction variability.

### 2.1 Wake Model

The impact of wakes on turbine power production is modeled using the FLORIS engineering wake modeling tool (NREL (2019)). Specifically, the Gaussian wake model developed by Bastankhah and Porté-Agel (2014, 2016) and Niayifar and Porté-
Agel (2016) is used to model the velocity deficits and wake profile. This model includes the ambient turbulence intensity (TI) as a parameter that helps determine the rate of wake recovery and the wake expansion. Wake deflection caused by yaw misalignment is modeled using the wake deflection model of Bastankhah and Porté-Agel (2016), based on the Reynolds-averaged Navier-Stokes equations. More information about the wake and wake deflection models available in FLORIS can be found in Annoni et al. (2018b) or at https://floris.readthedocs.io.
An example of the wakes produced by a two-turbine array with 5 rotor diameter ($D$) spacing using the above-mentioned wake model, with mean freestream wind speed $U$ = 8 m/s and a TI value of 10%, is provided in Fig. 1. The wake behavior is shown for the baseline case of zero yaw misalignment as well as with a positive $20°$ yaw offset applied to the upstream turbine. Note that a positive yaw offset corresponds to a counterclockwise rotation of the turbine relative to the wind direction.

### 2.2 Wind Turbine

The wake behavior and power production computed by FLORIS relies on a simplified wind turbine model, which is based on the NREL 5-MW reference wind turbine model in this research (Jonkman et al. (2009)). The NREL 5-MW reference turbine has a rotor diameter of 126 m and a hub height of 90 m. All analysis in this paper is based on simulations with a mean freestream wind speed of $U$ = 8 m/s, corresponding to a power production of 1.81 MW (rated wind speed for the NREL 5-MW reference model is 11.4 m/s). With $U$ = 8 m/s, the NREL 5-MW reference turbine operates in region 2, where power production
is maximized and thrust is relatively high (the coefficient of thrust $C_T$ = 0.762), conditions in which wake losses are high and wake steering is most effective.

To model the impact of yaw misalignment on power production, a simple cosine power law relationship is used in conjunction with the standard power equation in FLORIS:

$$P = \frac{1}{2}\rho A C_P u^3 \cos^p \gamma, \tag{1}$$

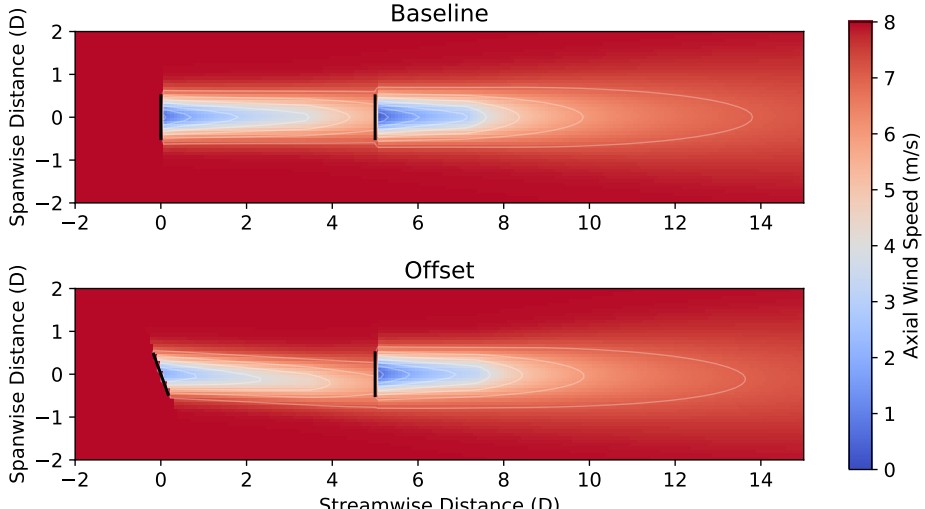

**Figure 1.** Examples of wakes for a two-turbine scenario with 5 rotor diameter ($D$) spacing using FLORIS. In the baseline case, both turbines are aligned with the wind direction. For the offset case, the upstream turbine has a yaw offset of $20°$.

where $\gamma$ is the yaw offset and the exponent $p$ describes how quickly power decreases with increasing yaw misalignment. A value of $p = 1.88$ is used here, based on fitting Equation 1 to data from LES simulations, as reported by Gebraad et al. (2016).

### 2.3    Yaw Controller

Yaw control is simulated using simple control logic based on the yaw controller model described by Bossanyi (2018). The
filtered wind direction determined by adding the relative wind direction measured by the wind vane to the nacelle position is compared to the raw nacelle position. When the magnitude of the difference exceeds a threshold of $8°$, the turbine begins yawing toward the direction of the filtered wind direction at the yaw rate of 0.3 °/s defined for the NREL 5-MW reference turbine (Jonkman et al. (2009)). Once the difference between the yaw position and the filtered wind direction reaches zero or changes sign, the turbine stops yawing until the error threshold is exceeded again.

### 2.4    Yaw Offset Controller

For a specific wind direction, optimal yaw offsets are found for the upstream turbine in a turbine pair by determining the offset that maximizes the sum of the power produced by the two turbines using FLORIS. Based on an analysis of the impact of yaw misalignment on turbine loads, which shows that certain loads are reduced with positive yaw misalignments but increase with negative yaw misalignments (Damiani et al. (2018)), only positive yaw offsets are considered here. Additionally, LES
simulations show that positive yaw misalignments are more effective at increasing power production as a result of the behavior of large-scale trailing vortices that help steer the wake, as explained by Fleming et al. (2018), as well as the impact of the



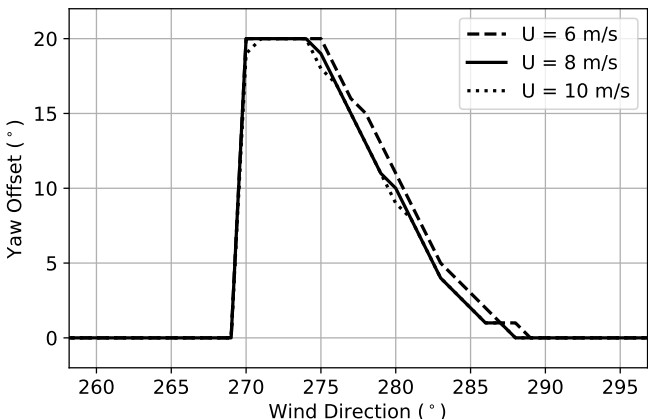

**Figure 2.** Yaw offset schedules optimized for a $5D$ turbine spacing with static wind directions for mean wind speeds 6, 8, and 10 m/s.

Coriolis force on wake deflection, discussed by Archer and Vasel-Be-Hagh (2019). To further reduce the impact of wake steering on turbine loads, yaw offsets are limited to 20° (Damiani et al. (2018)).

Yaw offsets for the upstream turbine in a two-turbine array aligned in the east-west direction, optimized for a turbine spacing of $5D$ with TI = 10%, are provided in Fig. 2 for mean wind speeds $U = 6$, 8, and 10 m/s. These "static-optimal" yaw offsets are
optimized assuming static wind directions (i.e., without wind direction variability). As the wind direction crosses above 270°, where the downstream turbine is fully waked, the highest allowable offset of 20° results in the maximum combined power production. As the wind direction increases to the north and the downstream turbine is increasingly only partially waked, the yaw offset needed to sufficiently deflect the wake decreases until there is no longer any benefit from wake steering.

Yaw offset control is used to apply the desired yaw offsets to a turbine and can either be implemented as direct yaw control,
wherein a direct yaw position command is sent to the turbine, or indirect yaw control, where the yaw error setpoint of the standard yaw controller is changed to the target offset. Although more precise yaw offset tracking can be achieved using direct yaw offset control (Bossanyi (2018)), indirect yaw offset control is considered in this research because it can be implemented in the field without modifying the turbine's yaw control logic (Fleming et al. (2019)).

The control logic used to implement indirect yaw offset control in this research, which is based on the strategy implemented
at a commercial wind farm by Fleming et al. (2019), is provided in Fig. 3. A modified wind vane signal is formed by subtracting the target yaw offset from the original wind vane signal. The modified vane signal is then fed into the wind turbine's standard yaw controller, causing it to track the target yaw offset instead of the default setpoint of zero. The target yaw offset is determined using a lookup table providing yaw offset as a function of nacelle-based wind speed and direction. Low-pass filtering is applied to the lookup table inputs to provide estimates of the slowly varying mean wind speed and direction. Note that for this study,
which considers below-rated operation for $U = 8$ m/s, the controller is simplified by containing only the yaw offset schedule determined for $U = 8$ m/s because of the low sensitivity of the optimal yaw offsets to wind speed variations in this region. After





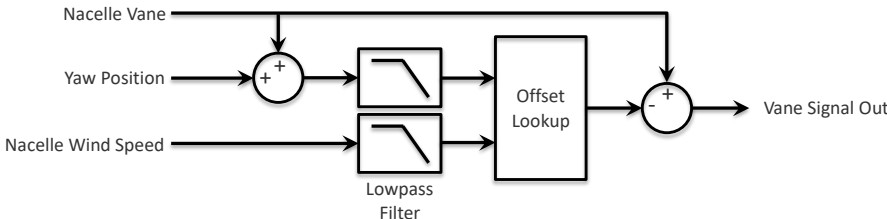

**Figure 3.** Yaw offset controller. Inputs include measurements from the wind turbine's nacelle vane, nacelle anemometer, and yaw position sensor. The output vane signal is used as the input to the wind turbine's yaw controller.

comparing the energy gain resulting from wake steering simulations with different wind direction filter time constants, a value of 30 s was chosen, yielding an estimate of the slowly varying wind direction without introducing too much delay.

## 2.5 Wind Model

To assess wake steering control with wind direction variability, we developed a dynamic wind model representing realistic wind conditions. Time series representing the turbulent wind direction at a point near hub height measured by the nacelle wind vane are needed to simulate the yaw and yaw offset controllers. FLORIS, on the other hand, models the time-averaged wake behavior and power production resulting from turbulent wind conditions as a function of mean wind direction. Additionally, high-frequency, small-scale components of wind direction incorrectly indicate the direction of wake travel. Therefore, a more appropriate input to FLORIS would be a signal representing the slowly varying, large-scale mean wind direction across the wind farm, with the turbulent component removed. To model these two different wind direction signals, stochastic time series are generated representing the slowly varying mean wind direction across the wind farm (the "low-frequency" component) as well as the purely turbulent wind direction component, corresponding to a fixed mean wind direction. The low-frequency component is used as the input to FLORIS, whereas the sum of the two signals acts as the input to the yaw and yaw offset controllers.

As discussed in Section 2.4, because wind speed variability around $U = 8$ m/s is not expected to significantly impact the effectiveness of wake steering, the wind model is simplified by assuming a fixed freestream wind speed of 8 m/s. However, wind speed variability likely has a greater impact on wake steering near rated wind speed, wherein the relationship between wind speed, power, and thrust is more nonlinear.

Stochastic wind direction signals are simulated by generating a normally distributed random time series based on the power spectra of the low-frequency and turbulent wind direction components, assuming the wind directions can be represented as Gaussian random processes. Specifically, a series of Fourier components at discrete frequencies containing uniformly distributed random phases is generated, with magnitudes determined by the desired power spectrum (Shinozuka and Deodatis (1991)). We then apply the inverse discrete Fourier transform to obtain the stochastic time series with a sample period of 1 s.

The power spectrum used to generate the turbulent wind direction component, $S_{\phi_t}(f)$, is based on data from LES using NREL's Simulator fOr Wind Farm Applications tool (Churchfield et al. (2012)), representing a neutral atmospheric boundary





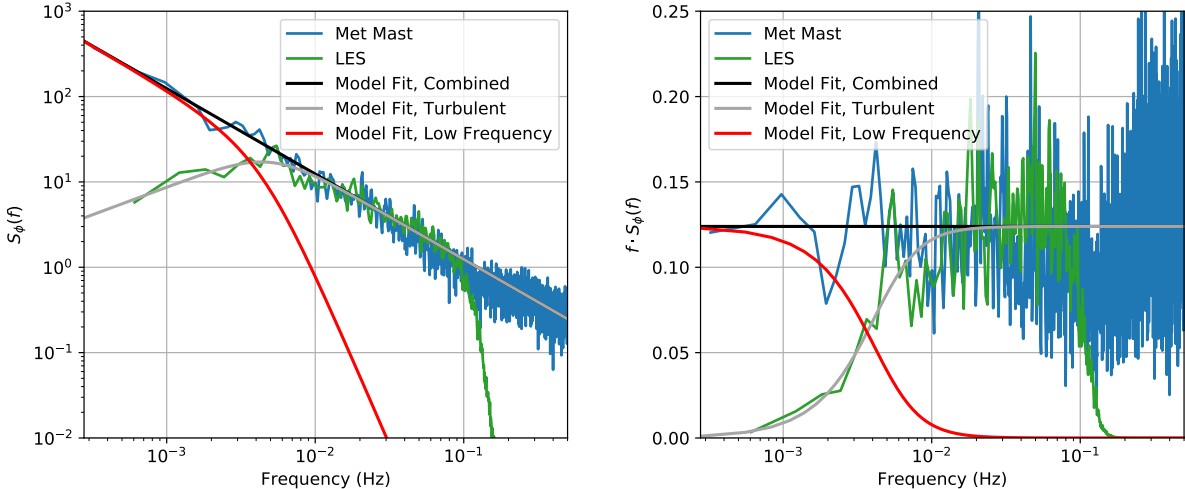

**Figure 4.** Normalized power spectra of the wind direction determined from met mast wind vane measurements, the turbulent wind direction component derived from LES, and the resulting low-frequency wind direction component (left); the same power spectra multiplied by frequency (right).

layer with a mean hub height wind speed of $U = {\sim}8$ m/s, which is similar to simulations discussed in past studies (Fleming et al. (2015, 2018)). The power spectrum is calculated using data at a height of 95 m with a mean wind speed of $U = 8.1$ m/s, TI = 10.1%, and a wind direction standard deviation of $3.93°$ (see the "LES" spectrum in Fig. 4).

Measurements obtained from a wind vane at a height of 87 m on the M5 meteorological (met) mast at NREL's National

Wind Technology Center (Clifton et al. (2013)) are used to determine the power spectrum of the combined low-frequency and turbulent wind directions, $S_\phi(f)$. To match the conditions generated by LES, data were limited to 1-hour periods, wherein the mean wind speed was between 7.5 m/s and 8.5 m/s and the atmospheric conditions were neutral (defined using the Monin-Obukhov stability parameter, $z/L$ (Clifton et al. (2013)), at a height of 15 m, as $|z/L| < 0.05$ (Rajewski et al. (2013))). Using twelve 1-hour periods of acceptable data with an average 1-hour TI = 18.8% and an average 1-hour wind direction standard

deviation of $10.92°$, a representative wind direction power spectrum is determined, shown as the "Met Mast" spectrum in Fig. 4.

Using the power spectra of the LES-based turbulent wind direction component, $S_{\phi_t}(f)$, and the met-mast-derived combined wind direction, $S_\phi(f)$, and assuming no correlation between the low-frequency and turbulent components, the spectrum of the low-frequency component, $S_{\phi_l}(f)$, is found via the relationship

$$S_\phi(f) = S_{\phi_l}(f) + S_{\phi_t}(f). \tag{2}$$

The LES-based turbulent and met-mast-based combined wind direction spectra are compared in Fig. 4, normalized so they converge at high frequencies. Note that the LES-derived spectrum quickly decays above 0.1 Hz because of the spatial filtering inherent in LES. These frequencies are ignored, however, and the trend observed between 0.01 and 0.1 Hz is assumed to





continue at higher frequencies. Finally, the wind direction power spectra are approximated using the following equations fit to the data, where $S_{\phi_l}(f)$ is calculated as $S_\phi(f) - S_{\phi_t}(f)$ and $C$ serves as a scaling constant:

$$S_\phi(f) = \frac{C}{f} \tag{3a}$$

$$S_{\phi_t}(f) = \frac{C \cdot 6.26 \cdot 10^3 \cdot f^{0.65}}{\left(1 + \left(\frac{f}{5 \cdot 10^{-3}}\right)^3\right)^{0.55}} \tag{3b}$$

$$S_{\phi_l}(f) = \frac{C\left(\left(1 + \left(\frac{f}{5 \cdot 10^{-3}}\right)^3\right)^{0.55} - 6.26 \cdot 10^3 \cdot f^{1.65}\right)}{f\left(1 + \left(\frac{f}{5 \cdot 10^{-3}}\right)^3\right)^{0.55}} \tag{3c}$$

Although Equation 3a is not physically realistic because it describes a power spectrum containing infinite energy across all frequencies, it fits the data well for the range of frequencies simulated. Note that below 0.0037 Hz, the low-frequency wind direction is the dominant component of the combined wind direction signal; for higher frequencies, the turbulent component dominates. Thus, the low-frequency wind direction component could be estimated by low-pass filtering a measured wind direction time series using a cutoff frequency of 0.0037 Hz.

Examples of the turbulent wind direction and combined wind direction time series from LES and met mast measurements, respectively, are shown in Fig. 5 (a). Note that the LES simulation is limited to 860 s, whereas the met mast measurements are analyzed in 1-hour blocks. Using the power spectra given by Equations 3b and 3c, examples of stochastic time series representing the turbulent and low-frequency wind direction components as well as the combined wind direction, formed by summing the turbulent and low-frequency components, are provided in Fig. 5 (b). The stochastic time series are scaled to match the combined standard deviation of $10.92°$ observed in the met mast data.

## 2.6 Simulation Procedure

As described in Section 2.5, the stochastic low-frequency wind direction component acts as the wind direction used in FLORIS, whereas the combined low-frequency and turbulent wind direction serves as the realistic noisy input to the yaw and yaw offset controllers. A fixed wind speed of 8 m/s is used as the input to both FLORIS and the yaw offset controller. Wake steering control is evaluated by generating a series of 1-hour stochastic wind direction time series with a combined wind direction standard deviation of $10.92°$ for a range of mean wind directions. To capture all wind directions wherein wake steering control could be active, unique 1-hour simulations are performed for mean wind directions between $200°$ and $340°$ in increments of $0.05°$, resulting in 2800 simulations for each control scenario examined. For comparison purposes, baseline yaw control is simulated in addition to wake steering control for each wind direction time series. The first 10 minutes of data resulting from each simulation are discarded to eliminate controller start-up transients.

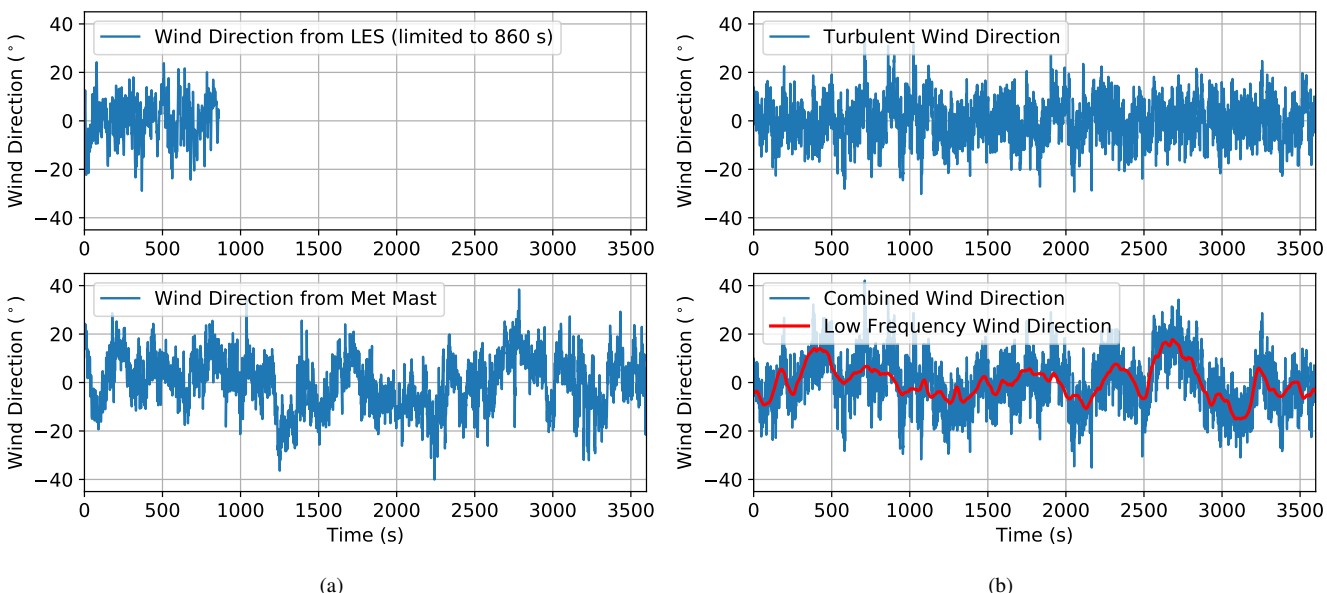

**Figure 5.** Example wind direction time series: (a) wind directions from LES and met mast wind vane measurements and (b) stochastic turbulent, low-frequency, and combined wind directions.

## 3 Yaw Offset Optimization

This section begins with a description of the methods used to model and quantify wind direction and yaw position uncertainty caused by wind direction variability. Next, the process for optimizing yaw offsets for wake steering control to maximize energy production with wind direction and yaw uncertainty is explained.

### 3.1 Wind Direction and Yaw Position Uncertainty

Wind direction and yaw position uncertainty are determined by first quantifying the yaw error variability. Standard yaw control is simulated using stochastic low-frequency and combined wind direction signals with the combined wind direction standard deviation of $10.92°$ measured in the field, as explained in Section 2.5. Yaw error variability is then quantified as the standard deviation of the difference between the low-frequency wind direction and the turbine's yaw position. An example of simulated wind direction and yaw position signals for both baseline yaw control and yaw offset control using static-optimal offsets is provided in Fig. 6. Baseline yaw control for the simulated conditions results in a yaw error standard deviation of $\sigma_\epsilon = 5.25°$.

The impact of wind direction and yaw position uncertainty on wake steering is analyzed by modeling wind direction ($\phi$) and yaw position ($\theta$) as jointly distributed random variables formed by adding wind direction and yaw uncertainty to the static wind direction and yaw position defined by the yaw offset schedule. The resulting joint probability density function (PDF) of wind direction and yaw position is given by the convolution of the static PDF, $f_{\Phi,\Theta,s}(\phi,\theta)$, based on the offset schedule, $\gamma(\phi)$,





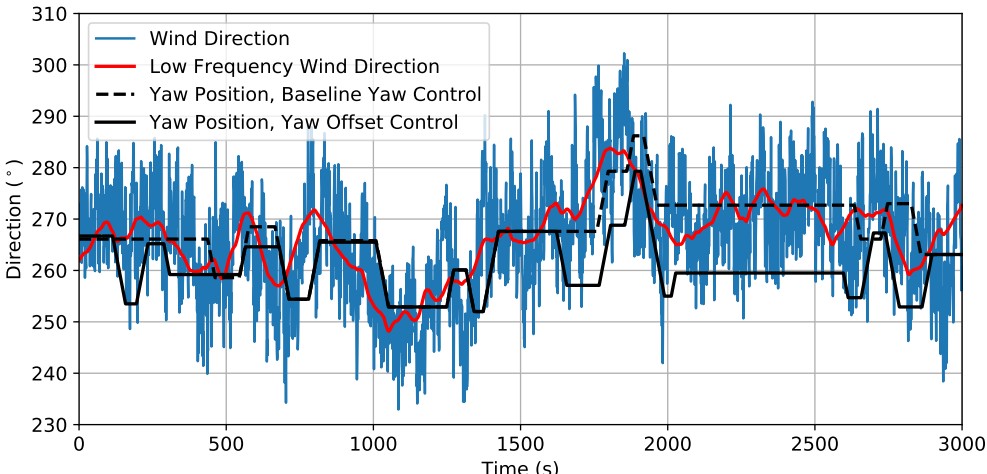

**Figure 6.** Example stochastic wind directions and low-frequency wind directions with yaw positions corresponding to baseline yaw control and yaw offset control, with the 8 m/s offset schedule shown in Fig. 2.

and the PDF representing the uncertainty in the two variables $f_{\Delta_\Phi, \Delta_\Theta}(\phi, \theta)$:

$$f_{\Phi, \Theta}(\phi, \theta) = f_{\Phi, \Theta, s}(\phi, \theta) * f_{\Delta_\Phi, \Delta_\Theta}(\phi, \theta). \tag{4}$$

The static PDF of wind direction and yaw position is given by

$$f_{\Phi, \Theta, s}(\phi, \theta) = f_\Phi(\phi) \delta(\theta - (\phi - \gamma(\phi))), \tag{5}$$

where $f_\Phi(\phi)$ is the PDF of the wind direction, assumed to be uniformly distributed across all wind directions to simplify the analysis. However, $f_\Phi(\phi)$ could easily be replaced by a more appropriate probability distribution based on site-specific conditions.

As explained by Rott et al. (2018), the PDF of the wind direction during a 5-minute time period can be approximated as a normal distribution. Because wind turbines typically yaw every few minutes or so—remaining at fixed yaw positions otherwise

(see Fig. 6)—the yaw error is also approximated as a normally distributed random variable. The yaw error uncertainty is then divided into wind direction uncertainty, $\Delta_\phi$, and yaw position uncertainty, $\Delta_\theta$, which are treated as independent normally distributed random variables described by the joint PDF:

$$f_{\Delta_\Phi, \Delta_\Theta}(\Delta_\phi, \Delta_\theta) \sim \mathcal{N}\left(\begin{bmatrix} 0 \\ 0 \end{bmatrix}, \begin{bmatrix} \sigma_\phi^2 & 0 \\ 0 & \sigma_\theta^2 \end{bmatrix}\right). \tag{6}$$

To maintain the observed yaw error standard deviation of $\sigma_\epsilon = 5.25°$, the following relationship between the variances of wind

direction uncertainty, yaw position uncertainty, and yaw error must exist:

$$\sigma_\phi^2 + \sigma_\theta^2 = \sigma_\epsilon^2. \tag{7}$$



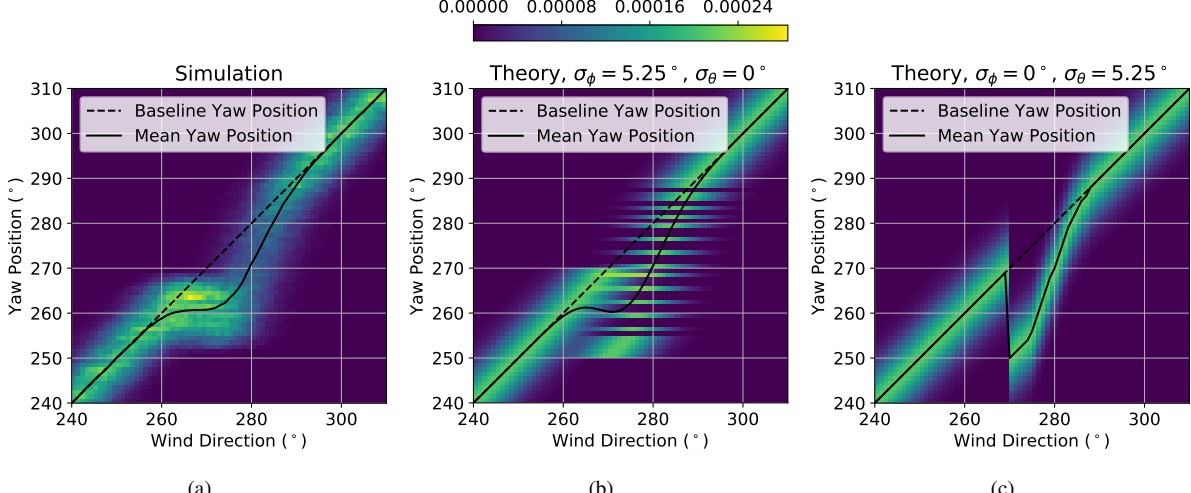

**Figure 7.** Distributions of wind direction and yaw position with yaw offset control using a static-optimal yaw offset schedule: (a) histogram from simulation results, and theoretical probability density functions assuming (b) only wind direction uncertainty, and (c) only yaw position uncertainty.

The impact of the parameters $\sigma_\phi$ and $\sigma_\theta$ on the PDF of wind direction and yaw position with wake steering based on the static-optimal offset schedule for 8 m/s described in Section 2.4 is shown in Fig. 7, which compares theoretical joint PDFs of wind direction and yaw position using Equations 4 through 7 with a histogram determined from simulation. All theoretical PDFs and histograms are discretized using 1° bins. The theoretical PDF of wind direction and yaw position assuming all of the

yaw error variation can be attributed to wind direction uncertainty ($\sigma_\phi = 5.25°$) is shown in Fig. 7b, whereas the PDF calculated assuming all variation is caused by yaw position uncertainty ($\sigma_\theta = 5.25°$) is provided in Fig. 7c. Also shown in the plots are the mean yaw positions achieved as a function of wind direction. Note that the two theoretical PDFs are identical for wind directions far from the yaw offset control sector; for baseline yaw control, any combination of wind direction and yaw position standard deviation that satisfies Equation 7 produces the same joint PDF of wind direction and yaw position. Consequently,

wind directions where wake steering is implemented must be used to identify the proper values of $\sigma_\phi$ and $\sigma_\theta$.

Rather than attributing all of the yaw error to either yaw position or wind direction uncertainty, Fig. 7 reveals that yaw offset control with wind direction variability is likely modeled best using a combination of the two sources of uncertainty. Assuming all of the yaw error is caused by wind direction uncertainty, as shown in Fig. 7b, implies that the yaw positions determined from the yaw offset schedule are achieved without any uncertainty. But once the yaw offsets are reached, the wind

direction varies while the yaw position is fixed, until the turbine yaws again. Note that the yaw position gaps in Fig. 7b are a consequence of discretizing the PDF using 1° bins. Alternatively, Fig. 7c highlights how attributing all of the yaw error to yaw position uncertainty suggests uncertainty in the yaw position that is achieved for a given wind direction, but that there is no wind direction variability once the yaw position is reached. The simulation-based histogram in Fig. 7a, however, exhibits





characteristics of both yaw position and wind direction uncertainty. Because the mean yaw positions achieved based on the simulation results are closer to the theoretical mean yaw values with no yaw position uncertainty, most of the yaw error can likely be attributed to wind direction uncertainty, as modeled by Rott et al. (2018). The procedure used to quantify the amount of wind direction and yaw position uncertainty, described by $\sigma_\phi$ and $\sigma_\theta$, respectively, used in the remainder of this research is

explained in Section 3.3.

### 3.2 Yaw Offset Optimization Procedure

For a given estimated low-frequency wind direction, $\hat{\phi}_l$, determined by the yaw offset controller, the optimal yaw offset $\gamma^*$ for the upstream turbine in the turbine pair is found using

$$\gamma^*\left(\hat{\phi}_l\right) = \underset{\gamma}{\mathrm{argmax}}\, E\left[P\left(\hat{\phi}_l, \gamma\right)\right], \tag{8}$$

where $E\left[P\left(\hat{\phi}_l, \gamma\right)\right]$ is the expected power production given the estimated wind direction, $\hat{\phi}_l$, and target yaw offset, $\gamma$. Based on the joint probability distribution of wind direction and yaw position uncertainty in Equation 6, the expected power production using FLORIS is given by

$$E\left[P\left(\hat{\phi}_l, \gamma\right)\right] = \int_{-180}^{180}\int_{-180}^{180} f_{\Delta_\Phi, \Delta_\Theta}\left(\Delta_\phi, \Delta_\theta\right) P_{FLORIS}\left(\hat{\phi}_l + \Delta_\phi, \gamma - \Delta_\theta\right) d\Delta_\phi d\Delta_\theta, \tag{9}$$

where $P_{FLORIS}\left(\phi, \gamma\right)$ describes the power production of the two-turbine array as a function of wind direction and the yaw

offset of the upstream turbine, and $\Delta_\phi$ and $\Delta_\theta$ represent deviations of the wind direction and yaw position from their mean values, respectively. Equations 8 and 9 essentially comprise the same form of optimization used by Quick et al. (2017), considering only yaw uncertainty, and Rott et al. (2018), examining wind direction uncertainty. Equation 9 is implemented by approximating the integration as a double summation and discretizing wind direction and yaw position using a step size of $1°$. As a result, only integer yaw offset values are considered.

The solution to Equations 8 and 9 as a function of wind direction without wind direction or yaw position uncertainty ($\sigma_\phi = \sigma_\theta = 0$) yields the "static-optimal" yaw offset schedule. When wind direction and yaw uncertainty, resulting from wind direction variability, are included, the solution is referred to as the "dynamic-optimal" offset schedule.

### 3.3 Wind Direction and Yaw Position Variability Parameter Tuning

Appropriate values for the standard deviation of the wind direction and yaw uncertainty are found by comparing the expected

mean energy production with wake steering based on theoretical PDFs of wind direction and yaw with simulation results. Specifically, the mean energy production across all wind directions is calculated for different combinations of $\sigma_\phi$ and $\sigma_\theta$ adhering to Equation 7. The combination that best predicts the mean energy resulting from wake steering simulations is used for finding the dynamic-optimal yaw offset schedule.

To find the uncertainty parameters that best predict the energy production with dynamic-optimal yaw offsets, an iterative pa-

rameter tuning approach is used. An initial guess is made, setting yaw position uncertainty equal to wind direction uncertainty.





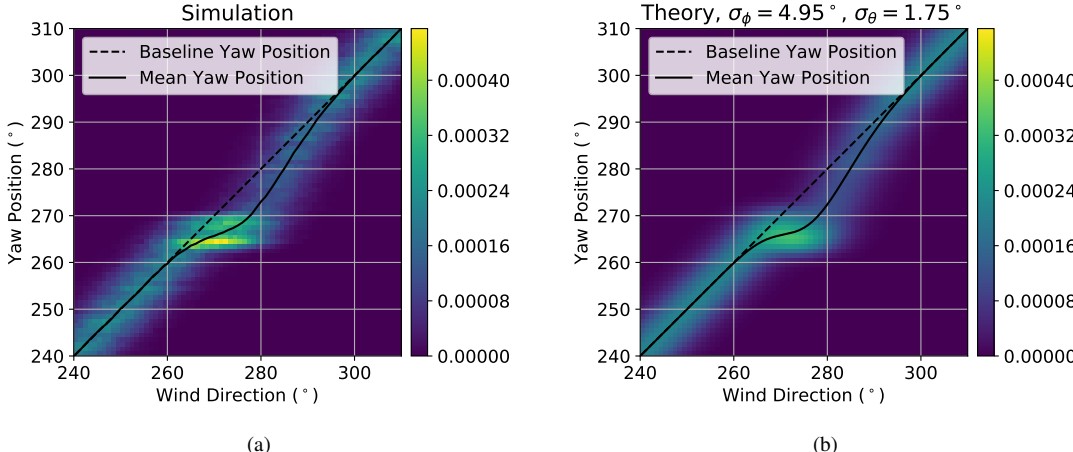

**Figure 8.** Distributions of wind direction and yaw position with yaw offset control using a dynamic-optimal yaw offset schedule: (a) histogram from simulation results and (b) theoretical probability density function assuming wind direction uncertainty with a standard deviation of 4.95° and yaw position uncertainty with a standard deviation of 1.75°.

Next, simulation results based on the initial dynamic-optimal yaw offset schedule are used to retune the uncertainty parameters. The process is repeated until both the uncertainty parameters and optimal yaw offsets converge. By applying this tuning procedure, values of $\sigma_\phi = 4.95°$ and $\sigma_\theta = 1.75°$ are found. As expected, most of the yaw error variation is attributed to wind direction uncertainty, with a small amount of yaw position uncertainty caused by the yaw controller dynamics. A comparison between the histogram of wind direction and yaw position based on wake steering simulations using the dynamic-optimal yaw offset schedule and the theoretical joint PDF with $\sigma_\phi = 4.95°$ and $\sigma_\theta = 1.75°$ is shown in Fig. 8.

A comparison between the histogram of wind direction and yaw position from wake steering simulations with the original static-optimal yaw offset schedule from Fig. 2 and the theoretical joint PDF using the tuned parameters is shown in Fig. 9. Although the mean yaw positions are similar, the wind direction and yaw distributions do not match as well as they do for the dynamic-optimal case, especially near $\phi = 270°$, where the wind direction is aligned with the turbine pair. Close agreement is desired so that the theoretical joint PDF of wind direction and yaw position can be used to accurately predict the simulated energy gain, allowing the optimal yaw offset schedule to be reliably found using the theoretical PDF. Part of this discrepancy can be explained by the larger offsets demanded by the static-optimal offset schedule. The theoretical joint PDF of wind direction and yaw assumes that the yaw controller tends to settle near one of the two yaw position extremes near $\phi = 270°$. However, the wake steering simulations reveal that the yaw controller often settles between the two extremes in this region as a result of the indirect yaw control implementation. Thus, the specific yaw control dynamics affect how well the theoretical joint PDF of wind direction and yaw predicts the actual control behavior. Notwithstanding the observed discrepancies for the static-optimal offset schedule, there is good agreement between the simulated and predicted joint PDFs for the robust dynamic-optimal case.





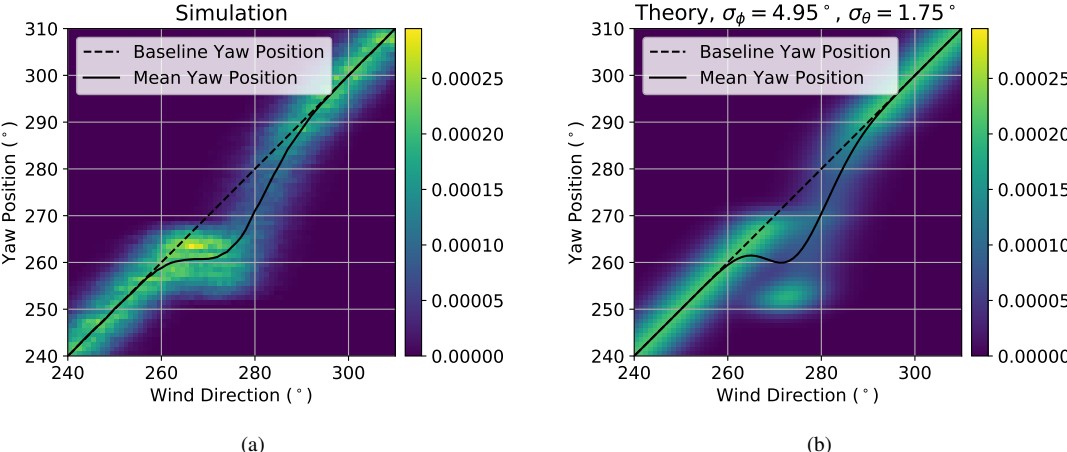

**Figure 9.** Distributions of wind direction and yaw position with yaw offset control using a static-optimal yaw offset schedule: (a) histogram from simulation results and (b) theoretical probability density function assuming wind direction uncertainty with a standard deviation of $4.95°$ and yaw position uncertainty with a standard deviation of $1.75°$.

## 4   Results

In this section, we present simulation results showing the increase in energy production with wake steering for a two-turbine array. In Section 4.1, the improvement in wake steering performance when dynamic-optimal yaw offsets are used is discussed in detail for a turbine spacing of $5D$ and turbulence intensity of $10\%$. The impact of turbine spacing and TI (which impacts wake expansion and recovery in FLORIS) on the improvement in energy production using dynamic-optimal yaw offsets is examined in Sections 4.2 and 4.3, respectively.

### 4.1   Comparison of Yaw Offset Controllers Optimized for Static and Variable Wind Directions

For a turbine spacing of $5D$ and turbulence intensity of $10\%$, the normalized mean power binned by wind direction for westerly wind directions is shown in Fig. 10 for baseline yaw control and wake steering control. The mean power production resulting from baseline yaw control and wake steering control with static-optimal as well as dynamic-optimal yaw offsets is provided in Fig. 10a for the case of static wind directions (i.e., power is computed for each wind direction independently, with yaw offsets determined directly from the yaw offset schedules). Clearly, the static-optimal offset schedule outperforms the lower-magnitude dynamic-optimal yaw offsets in this case. Note that power is only increased for one-half of the waked sector because of the restriction of positive yaw offsets. Figure 10b shows the mean power produced with the same three control scenarios with wind direction variability included. Solid lines correspond to theoretical predictions of power production based on Equation 9, whereas dashed lines are calculated from simulation results. Here, the static-optimal yaw offsets only outperform the dynamic-optimal offsets in a narrow sector; overall, the dynamic-optimal yaw offsets result in higher energy gain. Because of wind direction variability, the yaw offsets applied near the aligned direction cause a loss in power when the wind direction drops

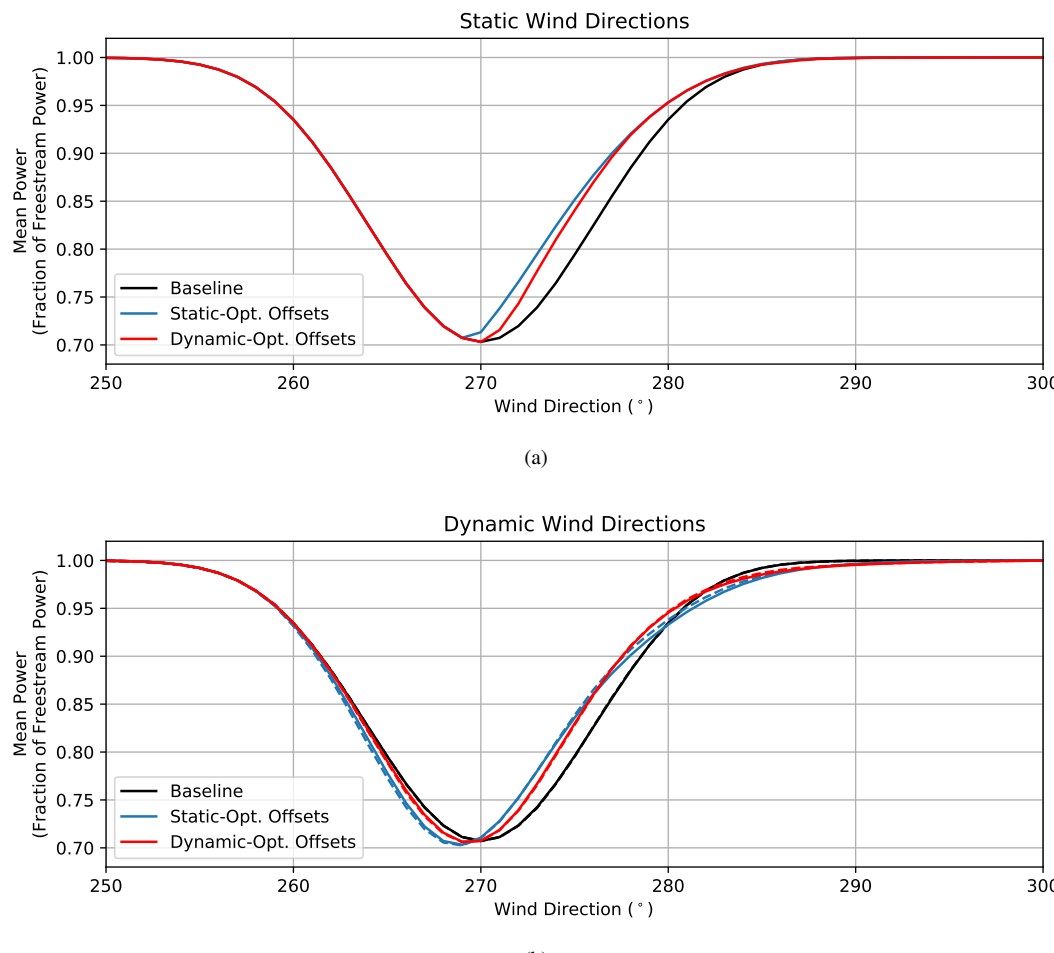

**Figure 10.** Normalized mean power for baseline yaw control and wake steering control with static-optimal and dynamic-optimal yaw offset schedules for (a) static and (b) dynamic wind directions. Mean power values are determined from theory (solid) and simulation (dashed).

below 270° or increases above ∼280°. Compared to the static-optimal offsets, the dynamic-optimal yaw offset schedule strikes a balance between achieving large gains between 270° and ∼280° and minimizing losses outside this sector.

Mean yaw offsets along with the normalized change in power with wake steering control are plotted in Fig. 11 for both static-optimal and dynamic-optimal yaw offsets. The mean achieved yaw offsets and changes in power with dynamic wind directions predicted by theory and resulting from simulation are shown along with the yaw offset schedules. Simulation results are provided based on binning using the original 1-s data as well as 1-minute and 10-minute averages of the data. The mean achieved yaw offsets binned by wind direction for the static-optimal and dynamic-optimal cases reveal the source of the power loss below 270° and above ∼280°, as shown in Fig. 10b. For wind directions below 270°, zero yaw offset is desired. However, the positive mean yaw offsets resulting from wind direction variability cause the wake to deflect clockwise toward the downstream turbine, reducing its power. On the other hand, for wind directions above ∼280°, relatively small yaw offsets





are needed to deflect the partial wake away from the downstream turbine. Because of wind direction variability, the resulting mean yaw offsets are too large, leading to unnecessary power loss on the upstream turbine with little additional gain at the downstream turbine. Comparing the static-optimal and dynamic-optimal cases shows how the less-aggressive dynamic-optimal yaw offsets lead to slightly lower peak gains in power production, but result in lower mean yaw offsets achieved outside of the

primary wake steering region, minimizing power loss. The lower peak gains are more than compensated for by the reduced losses. Finally, note that the dynamic-optimal yaw offsets extend to higher wind directions than the static-optimal offsets. The small offsets above 290° result in very little power loss, yet, due to wind direction variability, help increase the mean achieved yaw offsets at lower wind directions where wake steering is beneficial.

The theoretical predictions of the mean achieved yaw offsets and change in power with wake steering match the simulation

results very well for the dynamic-optimal yaw offset scenario in Figs. 11b and 11d, for which the wind direction and yaw position uncertainty model is tuned. For the static-optimal scenario, the theory predicts higher peak mean yaw offsets and a more narrow achieved yaw offset region than exhibited by the simulations. The predicted change in power also differs from the simulations more than in the dynamic-optimal case. As explained in Section 3.3, these discrepancies are related to yaw controller dynamics that are unaccounted for in the theoretical predictions. The overall gains for different wake steering sce-

narios are provided in Table 1, which lists the percentage of the wake losses recovered with dynamic wind directions assuming uniformly distributed wind directions. The baseline wake losses for the different scenarios investigated, from which the wake loss recovery is calculated, are provided in Table 2, again assuming uniformly-distributed wind directions. When simulating dynamic wind directions, 1.42% of the wake losses are recovered by the static-optimal offset schedule. The theoretical prediction of 1.08% slightly underestimates the energy gain because of unmodeled yaw controller dynamics, indicating room for

improvement in the theoretical model. By accounting for wind direction and yaw position uncertainty, wake steering simulations using dynamic-optimal yaw offsets result in an energy gain of 3.24% of the wake losses, nearly matching the predicted increase of 3.18% and representing an improvement of 128% over the static-optimal case.

Simulation results in Fig. 11 are binned using different averaging periods to illustrate how longer averaging periods can hide or smooth out some of the trends predicted by the theory. Averaging periods of 1 or 10 minutes are more relevant when

analyzing field data because of uncertainty in high-frequency wind direction measurements and to account for the time it takes for the wake to propagate between the pair of turbines. Although the mean achieved yaw offsets are not sensitive to these averaging periods, the mean change in power shows a strong dependence on the averaging period used for binning. As the binning period increases, the mean power trends are smoothed out. For example, using 10-minute averages, the simulation results for the dynamic-optimal scenario almost completely hide the power loss outside the main wake steering region, and

show a lower peak gain instead.

## 4.2   Impact of Turbine Spacing

For a two-turbine array, the turbine spacing impacts both the amount of energy that can be gained using wake steering and the additional benefit from using a dynamic-optimal yaw offset strategy. Keeping all other wind farm parameters the same as in Section 4.1, Fig. 12 compares the mean yaw offsets and change in power for both static-optimal and dynamic-optimal offset

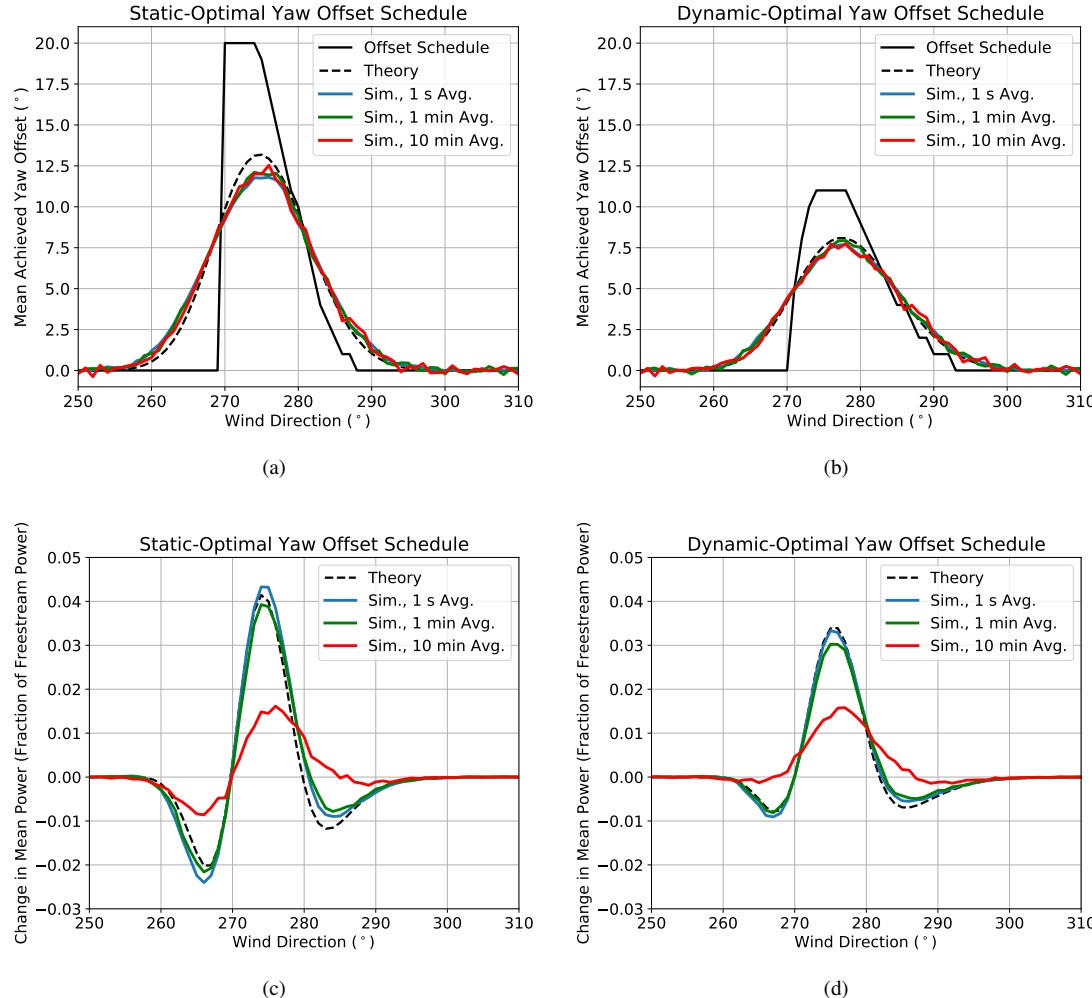

**Figure 11.** Mean achieved yaw offsets using offset schedules optimized for (a) static and (b) dynamic wind directions as well as change in mean power from wake steering using offset schedules optimized for (c) static and (d) dynamic wind directions. Theoretical achieved offsets and power gains are shown for dynamic wind directions along with the yaw offset schedules. Achieved yaw offsets and power gains from simulations with dynamic wind directions are provided for 1-s, 1-minute, and 10-minute averages of wind direction, yaw offset, and power.

schedules for turbine spacings of $3D$, $5D$, and $7D$, based on theory and simulation. Figure 12 shows that more energy can be gained using wake steering for shorter separation distances, where wake losses are higher (see Table 2). As the separation distance increases, baseline wake losses become lower, leaving less room for wake steering to improve energy production. However, as the separation distance increases, the wake steering losses suffered because of wind direction variability increase. As shown by the simulation results in Table 1, for a spacing of $3D$, wake steering control in dynamic wind conditions using the static-optimal offset schedule results in a wake loss recovery of 4.04%, whereas with the $7D$ spacing, energy is reduced



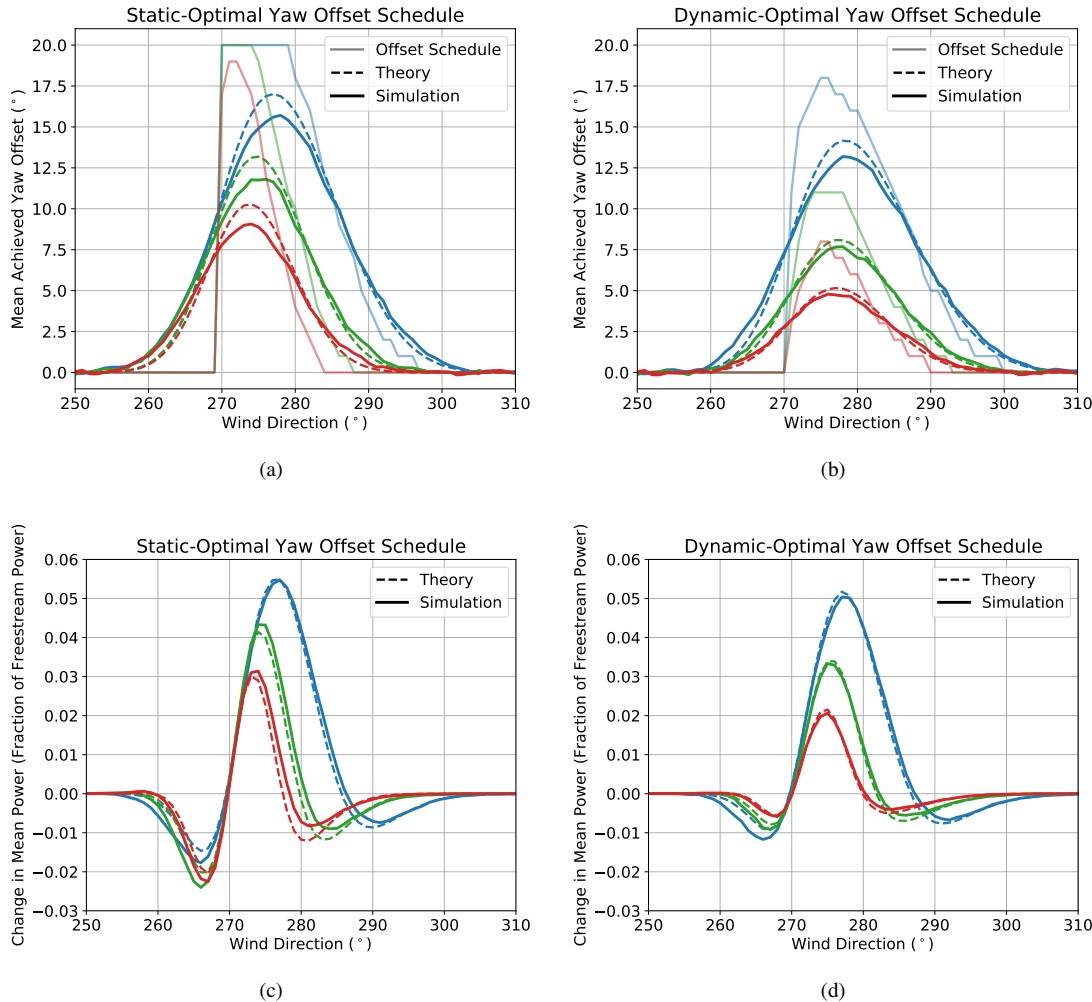

**Figure 12.** Mean achieved yaw offsets using offset schedules optimized for (a) static and (b) dynamic wind directions as well as change in mean power from wake steering using offset schedules optimized for (c) static and (d) dynamic wind directions for turbine spacings of $3D$ (blue), $5D$ (green), and $7D$ (red). Theoretical and simulation-based achieved offsets and power gains are shown for dynamic wind directions, along with the yaw offset schedules.

compared to the baseline case, yielding a wake loss recovery of -0.54%. Longer separation distances allow more time for a redirected wake to deflect by the time it reaches the downstream turbine. Consequently, unintended deviations from the optimal yaw offset result in larger changes to the wake position at the downstream turbine. For example, note the relatively high losses below 270° for the $7D$ spacing with static-optimal yaw offsets in Fig. 12c caused by mean positive yaw offset angles, which

5   significantly redirect the wake toward the downstream turbine.





Because the reduction in energy gain from wake steering caused by wind direction variability increases for larger turbine separations, the relative importance of using dynamic-optimal offset schedules increases as well. As shown in Figs. 12b and 12d, to achieve robustness to wind direction and yaw uncertainty, the peak dynamic-optimal offsets become smaller as the separation distance and, therefore, sensitivity to yaw offset deviations become greater. As a result, although the peak power gains are lower than with static-optimal yaw offsets, the losses below 270° and above the primary wake steering sector are greatly reduced. Referring to Table 1, for a separation of $3D$, replacing the static-optimal offsets with the dynamic-optimal offset schedule only raises the energy gained by wake steering from 4.04% of the total wake losses to 4.33%. But for a spacing of $7D$, switching to dynamic-optimal offsets changes the energy loss resulting from static-optimal offsets to an energy gain of 2.16% of total wake losses.

## 4.3 Impact of Turbulence Intensity

Just as wake steering can be implemented for a variety of turbine spacings, the effectiveness of wake steering depends on the atmospheric conditions. Although atmospheric stability has been shown to have a large impact on wake steering (Vollmer et al. (2016); Fleming et al. (2019)), ambient turbulence intensity (TI), which is closely linked to stability, acts as the primary atmospheric variable in the Gaussian wake model used in this research (Niayifar and Porté-Agel (2016)), affecting the degree of wake expansion and recovery. Turbulence causes the low-velocity air in the wake to mix with the surrounding higher-velocity flow, helping the wake recover. Additionally, turbulence causes wake meandering. Therefore, low TI leads to a narrow time-averaged wake profile with a high peak wake loss, whereas high turbulence causes a broader wake profile with lower peak losses. As listed in Table 2, the total wake losses decrease as the TI increases.

For a fixed turbine spacing of $5D$, Fig. 13 shows the offset schedules, mean achieved yaw offsets, and changes in power resulting from wake steering with both static-optimal and dynamic-optimal offsets for TI values of 5%, 10%, and 15%. The low-turbulence case with TI = 5% allows the greatest amount of energy to be gained using wake steering; with a narrow wake profile with deep losses, wake deflection causes a larger increase in velocity at the downstream turbine than for a more spread out wake profile with lower velocity deficits typical of high turbulence. At the same time, the higher sensitivity of the power production to changes in yaw offset also means that unintended yaw offsets can more easily steer the wake center back toward the downstream turbine, reducing power, as shown in Fig. 13c, between 260° and 270°. Because the greater susceptibility to wind direction and yaw position uncertainty outside of the primary wake steering control sector for lower TI values is somewhat balanced by the higher peak power gains inside the control sector, the relative impact of wind direction variability on the effectiveness of wake steering remains roughly constant as TI varies.

Similar to the results in Fig. 12, while the peak energy gains with dynamic-optimal offsets shown in Fig. 13d are lower than they are with static-optimal offset schedules, because of more robust, lower-magnitude offsets shown in Fig. 13b, the energy lost outside of the primary wake steering sector is greatly reduced. But just as the impact of wind direction variability on the overall effectiveness of wake steering does not strongly depend on TI, the relative improvement in the energy gain made possible by replacing static-optimal offset schedules with dynamic-optimal yaw offsets does not exhibit a clear trend with TI. The largest relative improvement in energy production after switching to dynamic-optimal yaw offsets occurs for the middle



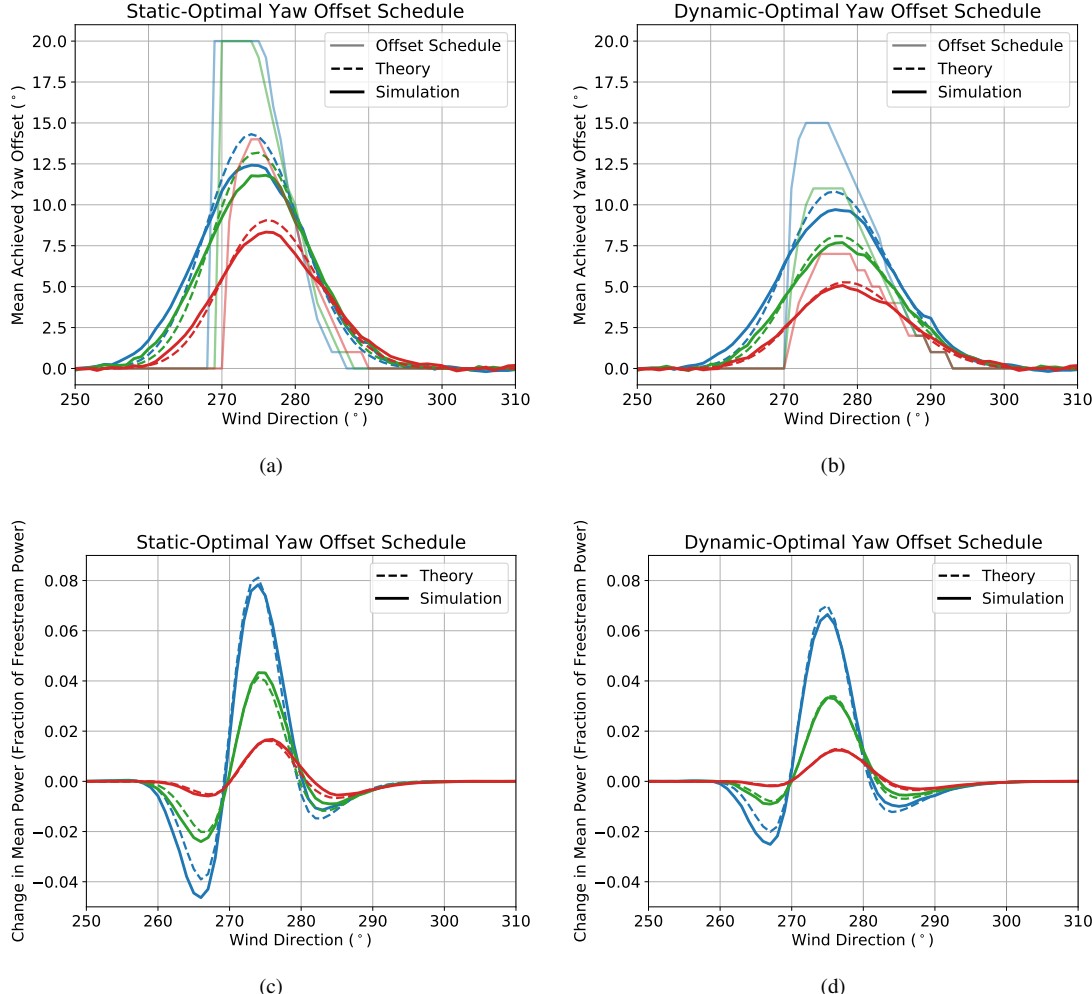

**Figure 13.** Mean achieved yaw offsets using offset schedules optimized for (a) static and (b) dynamic wind directions as well as change in mean power from wake steering using offset schedules optimized for (c) static and (d) dynamic wind directions for turbulence intensity values of 5% (blue), 10% (green), and 15% (red). Theoretical and simulation-based achieved offsets and power gains are shown for dynamic wind directions, along with the yaw offset schedules.

TI value of 10%, where the energy gain increases by 128%. However, large improvements in energy gains from wake steering are observed for both the lower and higher TI values as well (see Table 1). For TI = 5%, the energy gain increases by 54% from 2.94% of total wake losses to 4.53% when dynamic-optimal offsets are used. Similarly, after switching to the dynamic-optimal yaw offset schedule, the wake loss recovery for TI = 15% increases by 65% from 0.99% to 1.63%.





**Table 1.** Percentage of wake losses recovered from wake steering with dynamic wind directions for a two-turbine array with different turbine spacings and turbulence intensity values, assuming uniformly distributed wind directions. Results are provided for static-optimal and dynamic-optimal yaw offsets based on theoretical predictions as well as simulations.

| Yaw Offset Schedule/Simulation Case | Spacing (TI = 10%) | | | TI (Spacing = 5$D$) | | |
| --- | --- | --- | --- | --- | --- | --- |
| | 3$D$ | 5$D$ | 7$D$ | 5% | 10% | 15% |
| Static-Optimal/Theory | 4.17 | 1.08 | -1.67 | 3.67 | 1.08 | 0.54 |
| Static-Optimal/Simulation | 4.04 | 1.42 | -0.54 | 2.94 | 1.42 | 0.99 |
| Dynamic-Optimal/Theory | 4.47 | 3.18 | 2.04 | 5.18 | 3.18 | 1.56 |
| Dynamic-Optimal/Simulation | 4.33 | 3.24 | 2.16 | 4.53 | 3.24 | 1.63 |

**Table 2.** Total baseline wake losses compared to freestream operation for a two-turbine array with different turbine spacings and turbulence intensity values, assuming uniformly distributed wind directions.

| | Spacing (TI = 10%) | | | TI (Spacing = 5$D$) | | |
| --- | --- | --- | --- | --- | --- | --- |
| | 3$D$ | 5$D$ | 7$D$ | 5% | 10% | 15% |
| Wake Losses | 4.62% | 2.36% | 1.52% | 2.74% | 2.36% | 2.04% |

## 5 Discussion and Conclusions

This paper expanded on previous work investigating the optimization of wake steering control with yaw and wind direction uncertainty resulting from dynamic wind directions, particularly the research of Quick et al. (2017) and Rott et al. (2018). The present research examined the hypothesis that for steady-state wake models representing turbulent wind conditions, the most

relevant wind direction input should contain only the low-frequency wind direction component without the turbulence already captured by the wake model. This was accomplished by first comparing the power spectra of wind directions measured in the field and wind directions simulated using CFD for a fixed large-scale mean wind direction. Next, we generated stochastic time series representing the different dynamic wind direction components. Although previous work examined the impact of yaw uncertainty and wind direction uncertainty separately, here wake steering strategies were optimized for combined yaw

and wind direction uncertainty, estimated by comparing the yaw position resulting from realistic yaw and yaw offset control simulations with the low-frequency wind direction. However, it was found that wind direction uncertainty caused by wind direction variability, examined by Rott et al. (2018), is the dominant source of uncertainty.

For a two-turbine array, the theoretically predicted performance of wake steering control strategies optimized considering yaw and wind direction uncertainty was compared to results from realistic yaw offset control simulations, showing generally

good agreement. However, some discrepancies caused by unmodeled yaw controller dynamics exist. As discussed in Section 3.3, the discrepancies are related to the use of indirect yaw control; if the wind direction varies enough while the turbine is yawing, the yaw controller can stop yawing before the intended offset is reached. The agreement between theory and simulation could be improved by switching to direct yaw control, where exact yaw adjustments are prescribed by the controller.





An analysis of wake steering in dynamic wind conditions for different turbine spacings revealed that as the turbine separation increases, yaw and wind direction uncertainty has a more detrimental impact on the achievable gains in energy production. However, as the turbine spacing increases, the relative improvement in energy production when accounting for yaw and wind direction uncertainty in the yaw offset optimization process increases as well, as shown in Table 1. The impact of the degree

of wake expansion and recovery on wake steering with wind direction variability was examined by varying the turbulence intensity for a fixed turbine spacing. Unlike the dependence on turbine spacing, the relative improvements to wake steering that are achieved by considering wind direction and yaw position uncertainty in yaw offset optimization do not exhibit a strong relationship with turbulence intensity. The greatest improvement after switching to dynamic-optimal yaw offsets occurs for TI = 10%, the middle turbulence level that was investigated. Further research efforts are needed to determine how the amount

of wind direction variability depends on turbulence intensity as well as mean wind speed and atmospheric stability, beyond the 8 m/s, neutral stability case examined here.

Although not considered in this research, in addition to making wake steering more robust to wind direction and yaw position uncertainty, efforts can be made to reduce the uncertainty. For example, short-term forecasts of wind direction provided by remote-sensing instruments can be used by the wake steering controller to target more relevant future wind directions rather

than reacting to past measurements of the wind conditions. Another strategy for reducing uncertainty in the wind direction used by the controller is collective consensus control, discussed by Annoni et al. (2018a), where wind direction measurements from individual turbines in a wind farm are aggregated to form a more reliable wind direction estimate at each turbine. Collective consensus control can also be used to provide wind direction forecasts to downstream turbines.

Additionally, more realistic wake models are being developed based on new insights into the physics of wake steering that

may impact how susceptible wake steering is to wind direction variability. For example, the curled wake model, presented by Martínez-Tossas et al. (2019) and Bay et al. (2019), inspired by observations from CFD simulations discussed by Fleming et al. (2018), considers how trailing vortices resulting from yaw misalignment interact with the wake to not only deflect it but change its shape. Fleming et al. (2018) discuss how the trailing vortices created by multiple turbines can merge, creating large-scale structures in the flow that could potentially be used to entrain higher energy flow from above the wind farm. It is

likely that such "flow control" strategies are more robust to wind direction variability because they increase energy production over a large region of the wind farm rather than solely relying on deflecting individual wakes away from downstream turbines. However, future work is needed to investigate these more advanced wind farm control techniques and how they perform in dynamic wind conditions.

Finally, we investigated a two-turbine array in this study because it serves as a simple example for field validation. In follow-

on research, the yaw offsets and change in energy predicted by the models and simulations presented here will be compared to values obtained from wake steering field experiments, such as the campaign described by Fleming et al. (2019), to determine how accurately the proposed methods model wake steering in practice. As revealed in Section 4.1, longer averaging times used in the data analysis can hide some of the trends in the change of energy against wind direction predicted by the theory. Therefore, averaging times on the order of 1 minute should be considered for field validation.



*Acknowledgements.* This work was authored by the National Renewable Energy Laboratory, operated by Alliance for Sustainable Energy, LLC, for the U.S. Department of Energy (DOE) under Contract No. DE-AC36-08GO28308. Funding provided by the U.S. Department of Energy Office of Energy Efficiency and Renewable Energy Wind Energy Technologies Office. The views expressed in the article do not necessarily represent the views of the DOE or the U.S. Government. The U.S. Government retains and the publisher, by accepting the
5  article for publication, acknowledges that the U.S. Government retains a nonexclusive, paid-up, irrevocable, worldwide license to publish or reproduce the published form of this work, or allow others to do so, for U.S. Government purposes.



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
