# Peer review of "Design and Analysis of a Wake Steering Controller with Wind Direction Variability"

_Wind Energy Science, 2019_

## Referee Comment (RC1) · Anonymous Referee #1 · 26 Jul 2019

A good paper with some interesting theoretical treatment of directional uncertainty.

I only have one criticism, in the way the performance improvements are quantified, for example on page 17 line 22, where an improvement of 128% seems dramatic, but is actually only an increase in a change in wake losses. Those changes in wake losses are themselves small, around 1% to 3%, representing an even smaller change in actual energy production. Given the many other uncertainties, not least in the underlying wake model itself, these small changes could easily be 'lost in the noise' in real life. Tables 1 and 2 only report the actual changes, which is not so misleading, so I don't think the dramatic percentage changes in differences should be reported in the text either. Better still would be to report the percentage change in power prodution, rather than the percentage change in the wake losses, even if the numbers won't look as dramatic.

Another comment is about the use of wake steering in one direction only. The justification is that yawing in the other direction increases certain loads. However there are many reasons why it may still be worth steering in both directions (there is as yet no consensus on this point):

- not all loads increase; some will decrease, and they may be more important loads, depending on the turbine design drivers.

- even if loads increase on the yawing (upstream) turbines, this may be compensated by decreases in the same loads when the wind direction changes so that the turbine benefits from wake steering at other turbines.

- the increased loading may not happen if the turbine uses individual pitch control.

It would therefore be interesting in future to extend the analysis to include bi-directional yawing. This introduces additional practical difficulties because of the sudden reversal in desired yaw offset as the wind direction passes the turbine alignment direction. However, to study this properly, non-Gaussian direction changes, driven by synoptic weather patterns for example, may become important.

---

## Author Comment (AC1) · 20 Dec 2019

Thank you for your comments on the manuscript and for your interest in the topic. We plan to wait until all of the reviewers have provided comments before revising the manuscript and will respond again regarding the revisions. But in the meantime, we'd like to reply to your comments because it has been a while since you shared them with us.

1) A good paper with some interesting theoretical treatment of directional uncertainty.

We appreciate your interest.

2) I only have one criticism, in the way the performance improvements are quantified, for example on page 17 line 22, where an improvement of 128% seems dramatic, but

is actually only an increase in a change in wake losses. Those changes in wake losses are themselves small, around 1% to 3%, representing an even smaller change in actual energy production. Given the many other uncertainties, not least in the underlying wake model itself, these small changes could easily be 'lost in the noise' in real life. Tables 1 and 2 only report the actual changes, which is not so misleading, so I don't think the dramatic percentage changes in differences should be reported in the text either. Better still would be to report the percentage change in power prodution, rather than the percentage change in the wake losses, even if the numbers won't look as dramatic.

Author response: This is a good point, and we agree that statements like "an improvement of 128%" can be misleading, especially because the change in actual energy produced is small. We will update the presentation of the gain in energy capture in the revised draft, focusing on the absolute percent change in energy or wake losses recovered. The reasons we decided to present the improvements in terms of percentage of wake losses recovered are:

a) this resolves the issue of deciding which sector of wind directions to include when calculating the baseline energy and energy with wake steering. Otherwise the percentage change in energy would depend on which wind directions are included, which is a somewhat arbitrary choice.

b) the percentage change in energy capture with wake steering is relatively small for this two-turbine scenario, when considering the entire sector of wind directions. Because the main objective of wake steering is to improve energy capture when applied to an entire wind farm, we feel the improvement in energy gain for a two-turbine example is not as relevant as wake losses recovered. We expect the metric of percentage of wake losses recovered to be more consistent for different wind farm sizes and layouts (although it will still vary) and thus more meaningful. We think presenting both the percentage change in energy and the percentage of wake losses recovered will make sense.

3) Another comment is about the use of wake steering in one direction only. The justification is that yawing in the other direction increases certain loads. However there are many reasons why it may still be worth steering in both directions (there is as yet no consensus on this point): - not all loads increase; some will decrease, and they may be more important loads, depending on the turbine design drivers. - even if loads increase on the yawing (upstream) turbines, this may be compensated by decreases in the same loads when the wind direction changes so that the turbine benefits from wake steering at other turbines. - the increased loading may not happen if the turbine uses individual pitch control. It would therefore be interesting in future to extend the analysis to include bi-directional yawing. This introduces additional practical difficulties because of the sudden rever- sal in desired yaw offset as the wind direction passes the turbine alignment direction. However, to study this properly, non-Gaussian direction changes, driven by synoptic weather patterns for example, may become important.

Author response: Another motivation for considering only positive yaw misalignments is that positive yaw misalignments have been shown to increase power at the downstream turbine more than negative offsets through high fidelity modeling (e.g., the following references).

Archer, C. L. and Vasel-Be-Hagh, A.: Wake steering via yaw control in multi-turbine wind farms: Recommendations based on large-eddy simulation, Sustainable Energy Technologies and Assessments, 33, 34–43, doi:10.1016/j.seta.2019.03.002, 2019.

Fleming, P., Annoni, J., Churchfield, M., Martinez-Tossas, L. A., Gruchalla, K., Lawson, M., and Moriarty, P.: A simulation study demonstrating the importance of large-scale trailing vortices in wake steering, Wind Energy Science, 3, 243–255, doi:10.5194/wes-3-243-2018, 2018.

However, that is a good point that a concensus has not been reached on which directions of yaw offsets should be used, especially when considering the load benefits on downstream turbines. Research suggests that some loads might increase regardless

of the sign of the misalignment while others decrease regardless of the sign. One motivation from a loads perspective for using positive offsets alone comes from the indication that blade root bending moment loads will be reduced for positive offsets but will increase for negative offsets, as shown in:

Damiani, R., Dana, S., Annoni, J., Fleming, P., Roadman, J., van Dam, J., and Dykes, K.: Assessment of wind turbine component loads under yaw-offset conditions, Wind Energy Science, 3, 173–189, doi:10.5194/wes-3-173-2018, 2018.

The blade load trends combined with the higher power gains with positive offsets and the added complexity when switching between large positive and negative misalignments motivated us to consider only positive misalignments. Furthermore, only positive offsets are used in several recent wake steering field experiments, and we intended for the results of this study to aid in the analysis of field experiments.

Considering the points you raise, we will explain why we are focusing on positive offsets, but discuss how implementing both directions could be beneficial and that this is still an open area of research.

---

## Referee Comment (RC2) · Andreas Rott (Referee) · 6 Jan 2020

The authors' investigations bring together previous studies on the yaw uncertainty and the uncertainty in the wind direction as well as wind direction dynamics in a meaningful way. These are decisive factors for the full-scale application of Wake Steering. The manuscript is logically structured and written understandably and the graphics are well explained in most cases (see note below). Therefore this paper is a good contribution to the field of application (and uncertainties in the application) of Wake Steering, which, although wake steering has been intensively studied for more than 10 years, has not been treated very much so far.

Nevertheless, I have a few small comments and questions, which I would like to list

below:

RC1 page 1 line 16f: In the abstract 128 %, energy gain is announced. When reading the manuscript it becomes clear that this is the relative recovery of losses due to wake effects. The wording can be somewhat misleading here.

RC2 page 2 line 4f: The introduction begins with a description of wind farm control. Here one should be careful with this term. Wind farm control is usually something much more general, namely a power plant control, to comply with the grid codes. Yaw control usually belongs to the field of turbine control, and the advantages of coordinated yaw control on a wind farm level are only a "relatively" new subfield of wind farm control.

RC3 page 5 line 4f: Here it should be mentioned what kind of filter is used and, (if true) that it is also described in Bossanyi (2018).

RC4 page 5 line 5f: Why are you comparing the wind vane signal plus the nacelle position to the nacelle position and not just use the wind vane signal?

RC5 page 7 line 25: The acronym SOWFA is quite well known in the community and should be mentioned here.

RC6 page 11 line 4f: To avoid confusion please mention, that \delta is the Kronecker-Delta.

RC7 page12 line 1f: Which wind direction signal was used in the joint distribution in Fig. 7. The low-filtered wind direction or the "combined" wind direction?

RC8 page 13 Equation (9) For easier readability in the equations, I would advise to only italicize variables, as the ISO standard suggests. The l (in \hat\phi_l), FLORIS and the d of the integrator should be written in roman.

RC9 page 13 Section 3.3: Here the parameters for the uncertainty in the wind direction (x-axis Fig.8) and the yaw (y-axis Fig. 8) are tuned using the simulation. But in the simulation, the yaw uncertainty should either not exist or be adjustable. Can you explain

the result for the yaw uncertainty? Is it possible that the hysteresis of the Yaw controller and the Yaw process used in the simulation influences the parameterization?

RC10 page 16 Figure 10: The normalized power shown here is probably the sum of both turbines and not just the turbine downstream. This should be clearly stated.

---

## Author Comment (AC2) · 21 Jan 2020

Dear Andreas,

Thank you for your comments and questions about our submitted manuscript and for your interest in this work. You have brought up some good points, and below you will find our response to your comments:

RC1 page 1 line 16f: In the abstract 128 %, energy gain is announced. When reading the manuscript it becomes clear that this is the relative recovery of losses due to wake effects. The wording can be somewhat misleading here.

Author response: Good point, and we have received this feedback from the other referee as well. In the revised manuscript we plan to re-word this to emphasize the ab-

solute change in the energy gain when comparing the "dynamic-optimal" and "static-optimal" yaw offsets (e.g., x% of wake losses recovered compared to y%), rather than a potentially-misleading percentage of a percentage change.

RC2 page 2 line 4f: The introduction begins with a description of wind farm control. Here one should be careful with this term. Wind farm control is usually something much more general, namely a power plant control, to comply with the grid codes. Yaw control usually belongs to the field of turbine control, and the advantages of coordinated yaw control on a wind farm level are only a "relatively" new subfield of wind farm control.

Author response: We will modify the phrase to emphasize that we are describing the subset of wind farm control for active wake control.

RC3 page 5 line 4f: Here it should be mentioned what kind of filter is used and, (if true) that it is also described in Bossanyi (2018).

Author response: We use a 1st order low pass filter with a time constant of 35 seconds. In Bossanyi (2018), a time constant of 30 seconds is used. We will include a description of the filter and a comparison with the Bossanyi filter in the revised manuscript.

RC4 page 5 line 5f: Why are you comparing the wind vane signal plus the nacelle position to the nacelle position and not just use the wind vane signal?

Author response: Although using the filtered wind vane signal would be fine for determining when the error threshold has been passed to initiate yawing, comparing the raw yaw position to the filtered sum of yaw position and wind vane helps in determining when to stop yawing. If only the filtered wind vane were used to determine when the yaw error reaches zero and the turbine should stop yawing, it would take too long to reach zero yaw error, due to the filter delay, and the controller would overshoot the desired yaw position. Comparing filtered absolute wind direction to the raw yaw position more accurately reflects when the yaw position becomes aligned with the low frequency wind direction.

RC5 page 7 line 25: The acronym SOWFA is quite well known in the community and should be mentioned here.

Author response: We'll change this to SOWFA (Simulator fOr Wind Farm Applications).

RC6 page 11 line 4f: To avoid confusion please mention, that \delta is the Kronecker-Delta.

Author response: Good point, the delta function should have been introduced here.

RC7 page12 line 1f: Which wind direction signal was used in the joint distribution in Fig. 7. The low-filtered wind direction or the "combined" wind direction?

Author response: Starting in Fig. 7, the wind direction being plotted is the low frequency wind direction, which we expect to be more relevant as an input to FLORIS. We will clarify that this is the low frequency wind direction in the revised manuscript.

RC8 page 13 Equation (9) For easier readability in the equations, I would advise to only italicize variables, as the ISO standard suggests. The l (in \hat\phi_l), FLORIS and the d of the integrator should be written in roman.

Author response: Thank you for the suggestion. We will incorporate this feedback.

RC9 page 13 Section 3.3: Here the parameters for the uncertainty in the wind direction (x-axis Fig.8) and the yaw (y-axis Fig. 8) are tuned using the simulation. But in the simulation, the yaw uncertainty should either not exist or be adjustable. Can you explain the result for the yaw uncertainty? Is it possible that the hysteresis of the Yaw controller and the Yaw process used in the simulation influences the parameterization?

Author response: It is true that there is no yaw uncertainty in the simulations, in the sense that we assume the yaw position reported by the controller is the true yaw orientation. Therefore, the yaw "uncertainty" we are modeling is due to the yaw controller possibly stopping at a different yaw position than was originally intended when the yaw maneuver begins. For example, as the yaw controller is in the process of yawing to

achieve a 20 degree offset determined by the lookup table, the wind direction could change enough so that the target offset switches to zero degrees. The controller could then stop part way toward the original 20 degree offset because it is now overshooting the new 0 degree offset target. While this doesn't mean there's any uncertainty in the yaw position measurement, it acts as uncertainty in the achieved yaw position as a function of wind direction compared to the intended static yaw position curve. Another example is if the yaw controller is still trying to track a target yaw offset of zero after the wind direction has shifted to the wake steering sector, because the wind direction filter in the wake steering controller hasn't yet reached the new wind direction as a result of filter delay. Some of this yaw "uncertainty" could be removed by using direct yaw control. However, we wanted to show the performance of a standard baseline yaw controller implementing wake steering. In the revised manuscript, we will plan to elaborate on the sources of yaw uncertainty.

RC10 page 16 Figure 10: The normalized power shown here is probably the sum of both turbines and not just the turbine downstream. This should be clearly stated.

Author comment: Yes, we will clarify that it is the sum of the upstream and downstream turbine powers.

---

## Author Response (AR1)

**Author Response to reviewer 1**

Dear Reviewer,

Thank you for your review of this manuscript and for your interest in the topic. We have revised the manuscript based on your and the other reviewer's comments. You will find our responses to your comments below.

> A good paper with some interesting theoretical treatment of directional uncertainty.

We appreciate your interest in this research.

> I only have one criticism, in the way the performance improvements are quantified, for example on page 17 line 22, where an improvement of 128% seems dramatic, but is actually only an increase in a change in wake losses. Those changes in wake losses are themselves small, around 1% to 3%, representing an even smaller change in actual energy production. Given the many other uncertainties, not least in the underlying wake model itself, these small changes could easily be 'lost in the noise' in real life. Tables 1 and 2 only report the actual changes, which is not so misleading, so I don't think the dramatic percentage changes in differences should be reported in the text either. Better still would be to report the percentage change in power prodution, rather than the percentage change in the wake losses, even if the numbers won't look as dramatic.

This is a good point, and we agree that statements like "an improvement of 128%" were misleading in the original manuscript. Therefore, we have removed this statement from the abstract and revised the presentation of the results to emphasize the absolute change in the percentage of wake losses recovered rather than a potentially misleading large percentage change of a small percentage gain.

The reasons we decided to present the improvements in terms of percentage of wake losses recovered are:

a) this resolves the issue of deciding which sector of wind directions to include when calculating the baseline energy and energy with wake steering. Otherwise the percentage change in energy would depend on which wind directions are included, which is a somewhat arbitrary choice. For example, the energy gain could be calculated averaged over all wind directions, or could be averaged over only the waked sector, which itself could be defined in different ways.

b) the percentage change in energy capture with wake steering is relatively small for this two-turbine scenario, when considering the entire sector of wind directions. Because the main objective of wake steering is to improve energy capture when applied to an entire wind farm, we feel the absolute energy gain for a two-turbine example is not as relevant as wake losses

recovered. We expect the metric of percentage of wake losses recovered to be more consistent for different wind farm sizes and layouts (although it will still vary) and thus more meaningful.

In the 2nd to last paragraph of Section 4.1 on pg. 19, we have added the following explanation for expressing the energy gain from wake steering as a wake loss percentage: "Note that when averaging over all wind directions, the gain in absolute energy production from wake steering is small for this two-turbine scenario (≤ 0.2%), primarily because of the low baseline wake losses. Therefore, we express the gains as a percentage of wake losses recovered, which we expect to be a more meaningful value for comparison across different wind farm scenarios."

The presentation of the qualitative improvements in wake steering with dynamic-optimal yaw offsets in the text have been changed to try to avoid potentially misleading claims, and are summarized here. We left the percentage change in the gains in certain places to compare the relative benefit of wake steering with dynamic-optimal yaw offsets compared to static-optimal offsets, but emphasized that these are only relative improvements compared to wake steering with static-optimal yaw offsets.

- Abstract, pg. 1, ln. 15: "For a spacing of 5 rotor diameters and a turbulence intensity of 10%, robust yaw offsets optimized for variable wind directions yielded an energy gain *equivalent to 3.24% of wake losses recovered, compared to 1.42% of wake losses recovered with yaw offsets optimized for static wind directions*."

- Section 4.1, pg. 19, ln. 10: "When simulating dynamic wind directions, 1.42% of the wake losses are recovered by the static-optimal offset schedule. The theoretical prediction of 1.08% slightly underestimates the energy gain because of unmodeled yaw controller dynamics, indicating room for improvement in the theoretical model. By accounting for wind direction and yaw position uncertainty, wake steering simulations using dynamic-optimal yaw offsets result in an energy gain of 3.24% of total wake losses, nearly matching the predicted increase of 3.18% and representing *more than a twofold improvement over the static-optimal wake steering case*."

- Section 4.3, pg. 22, ln. 1: "The largest relative improvement in energy production after switching to dynamic-optimal yaw offsets occurs for the middle TI value of 10%, *where the percentage of wake losses recovered by wake steering increases from 1.42% to 3.24% (a relative improvement of 128%).* However, large improvements in energy gains from wake steering are observed for both the lower and higher TI values as well (see Table 1). *For TI = 5%, the energy gain increases from 2.94% of total wake losses to 4.53% when dynamic-optimal offsets are used (a relative change of 54%).* Similarly, after switching to the dynamic-optimal yaw offset schedule, *the wake loss recovery for TI = 15% increases from 0.99% to 1.63% (a relative change of 65%)*."

Another comment is about the use of wake steering in one direction only. The justification is that yawing in the other direction increases certain loads. However there are many reasons why it may still be worth steering in both directions (there is as yet no consensus on this point):

> - not all loads increase; some will decrease, and they may be more important loads, depending on the turbine design drivers.
>
> - even if loads increase on the yawing (upstream) turbines, this may be compensated by decreases in the same loads when the wind direction changes so that the turbine benefits from wake steering at other turbines.
>
> - the increased loading may not happen if the turbine uses individual pitch control.
>
> It would therefore be interesting in future to extend the analysis to include bi-directional yawing. This introduces additional practical difficulties because of the sudden reversal in desired yaw offset as the wind direction passes the turbine alignment direction. However, to study this properly, non-Gaussian direction changes, driven by synoptic weather patterns for example, may become important.

That is a good point that a consensus has not been reached on which directions of yaw offsets should be used, especially when considering the load benefits on downstream turbines. And we agree that our initial explanation for choosing only positive offsets perhaps made it sound like this was a settled issue. In reality it is more complicated. Research suggests that some loads might increase regardless of the sign of the misalignment while others decrease regardless of the sign, as you mention. One motivation from a loads perspective for using positive offsets alone comes from the indication that blade root bending moment loads will be reduced for positive offsets but will increase for negative offsets, as shown in:

Damiani, R., Dana, S., Annoni, J., Fleming, P., Roadman, J., van Dam, J., and Dykes, K.: Assessment of wind turbine component loads under yaw-offset conditions, Wind Energy Science, 3, 173–189, doi:10.5194/wes-3-173-2018, 2018.

The blade load trends combined with the higher power gains with positive offsets and the added complexity when switching between large positive and negative misalignments motivated us to consider only positive misalignments. Furthermore, only positive offsets are used in several recent wake steering field experiments, and we intended for the results of this study to aid in the analysis of field experiments. To address the points you raise, we have heavily revised the first paragraph of Section 2.4, provided below:

"For a specific wind direction, optimal yaw offsets are found for the upstream turbine in a turbine pair by determining the offset that maximizes the sum of the power produced by the two turbines using FLORIS. *Because of the practical challenges of switching between large positive and negative yaw offsets for small changes in wind direction as well as the relative benefits of positive yaw misalignments, only positive offsets are considered here. For example, Damiani et al. (2018) show that blade root bending moment fatigue is reduced with positive yaw misalignments but increases with negative yaw misalignments.* Additionally, LES

simulations show that positive yaw misalignments are more effective at increasing power production as a result of the behavior of large-scale trailing vortices that help steer the wake, as explained by Fleming et al. (2018), as well as the impact of the Coriolis force on wake deflection, discussed by Archer and Vasel-Be-Hagh (2019). To further reduce the impact of wake steering on turbine loads, we limit yaw offsets to 20° (Damiani et al. (2018)). *However, wake steering with both positive and negative yaw misalignments may be a promising strategy because of the additional energy that can be captured. Whereas research suggests that blade root bending moment fatigue decreases only for positive yaw offsets, loads for other components may increase or decrease regardless of the direction of misalignment (Damiani et al. (2018); Mendez Reyes et al. (2019)). Therefore, the specific design-driving loads should be identified and considered when assessing a wake steering strategy. Furthermore, the load reduction experienced by downstream turbines from wake steering could outweigh the higher loads on misaligned upstream turbines when averaged over the lifetime of the wind farm, as discussed by Kanev et al. (2018) and Mendez Reyes et al. (2019)."*

**Author Response to Reviewer 2 (Andreas Rott)**

Dear Andreas,

Thank you for your review of this manuscript and for your continued interest in this topic. We have revised the manuscript based on your and the other reviewer's comments. Please find our responses to your comments below.

> The authors' investigations bring together previous studies on the yaw uncertainty and the uncertainty in the wind direction as well as wind direction dynamics in a meaningful way. These are decisive factors for the full-scale application of Wake Steering. The manuscript is logically structured and written understandably and the graphics are well explained in most cases (see note below). Therefore this paper is a good contribution to the field of application (and uncertainties in the application) of Wake Steering, which, although wake steering has been intensively studied for more than 10 years, has not been treated very much so far. Nevertheless, I have a few small comments and questions, which I would like to list below:

We appreciate your interest in this research.

> RC1 page 1 line 16f: In the abstract 128 %, energy gain is announced. When reading the manuscript it becomes clear that this is the relative recovery of losses due to wake effects. The wording can be somewhat misleading here.

Thanks for this feedback, which the other reviewer noted too. We agree that this is a misleading statement. The presentation of the qualitative improvements in wake steering with dynamic-optimal yaw offsets in the text have been changed to try to avoid potentially misleading claims, and are summarized here. We left the percentage change in the gains in certain places to compare the relative benefit of wake steering with dynamic-optimal yaw offsets compared to static-optimal offsets, but emphasized that these are only relative improvements compared to wake steering with static-optimal yaw offsets.

- Abstract, pg. 1, ln. 15: "For a spacing of 5 rotor diameters and a turbulence intensity of 10%, robust yaw offsets optimized for variable wind directions yielded an energy gain *equivalent to 3.24% of wake losses recovered, compared to 1.42% of wake losses recovered with yaw offsets optimized for static wind directions*."

- Section 4.1, pg. 19, ln. 10: "When simulating dynamic wind directions, 1.42% of the wake losses are recovered by the static-optimal offset schedule. The theoretical prediction of 1.08% slightly underestimates the energy gain because of unmodeled yaw controller dynamics, indicating room for improvement in the theoretical model. By accounting for wind direction and yaw position uncertainty, wake steering simulations using dynamic-optimal yaw offsets result in an energy gain of 3.24% of total wake

losses, nearly matching the predicted increase of 3.18% and representing *more than a twofold improvement over the static-optimal wake steering case*."

- Section 4.3, pg. 22, ln. 1: "The largest relative improvement in energy production after switching to dynamic-optimal yaw offsets occurs for the middle TI value of 10%, *where the percentage of wake losses recovered by wake steering increases from 1.42% to 3.24% (a relative improvement of 128%).* However, large improvements in energy gains from wake steering are observed for both the lower and higher TI values as well (see Table 1). *For TI = 5%, the energy gain increases from 2.94% of total wake losses to 4.53% when dynamic-optimal offsets are used (a relative change of 54%).* Similarly, after switching to the dynamic-optimal yaw offset schedule, *the wake loss recovery for TI = 15% increases from 0.99% to 1.63% (a relative change of 65%).*"

> RC2 page 2 line 4f: The introduction begins with a description of wind farm control. Here one should be careful with this term. Wind farm control is usually something much more general, namely a power plant control, to comply with the grid codes. Yaw control usually belongs to the field of turbine control, and the advantages of coordinated yaw control on a wind farm level are only a "relatively" new subfield of wind farm control.

This is a great point, and we agree that wake control for increasing power or reducing loads is only a subfield of wind farm control. The beginning of the introduction section has been revised to more accurately reflect this:

"A subset of wind farm control strategies involves the control of individual wind turbines to influence the aerodynamic interaction between turbines in a wind farm via their wakes. These control strategies can improve the total energy production of a wind farm or reduce structural loads (Johnson and Thomas (2009); Boersma et al. (2017)). Although several methods of actuation exist for influencing the wake behind a wind turbine…"

> RC3 page 5 line 4f: Here it should be mentioned what kind of filter is used and, (if true) that it is also described in Bossanyi (2018).

We use a 1st order low pass filter with a time constant of 35 seconds. In Bossanyi (2018), a time constant of 30 seconds is used. The type of filter, the differences between our yaw controller and the one described by Bossanyi, and the reason for the slight difference are now explained in the heavily revised section 2.3 paragraph:

"Yaw control is simulated using simple logic based on the yaw controller model described by Bossanyi (2018). A slowly-varying wind direction signal is formed by low-pass filtering the measured wind direction, given by the sum of the wind vane signal and the nacelle position, using a first-order filter with a time constant of 35 s. When the magnitude of the difference between the filtered wind direction and the nacelle position exceeds a threshold of 8°, the turbine begins yawing toward the direction of the filtered wind direction at the yaw rate of 0.3

°/s defined for the NREL 5-MW reference turbine (Jonkman et al. (2009)). Once the difference between the current yaw position and the slowly-varying filtered wind direction reaches zero or changes sign, the turbine stops yawing until the error threshold is exceeded again. The values of the error threshold and yaw rate parameters used here are equivalent to those presented by Bossanyi (2018). However, instead of the 30 s filter time constant described by Bossanyi (2018), a slightly longer time constant of 35 s is used here, which results in yaw activity similar to that of a commercial wind turbine used in the wake steering experiment discussed by Fleming et al. (2019)."
* * *
RC4 page 5 line 5f: Why are you comparing the wind vane signal plus the nacelle position to the nacelle position and not just use the wind vane signal?
* * *
Although using the filtered wind vane signal would be fine for determining when the error threshold has been passed to initiate yawing, comparing the raw yaw position to the filtered sum of yaw position and wind vane helps in determining when to stop yawing. If only the filtered wind vane were used to determine when the yaw error reaches zero and the turbine should stop yawing, it would take too long to reach zero yaw error, due to the filter delay, and the controller would overshoot the desired yaw position. Comparing filtered absolute wind direction to the raw yaw position more accurately reflects when the yaw position becomes aligned with the low frequency wind direction. In Section 2.3, we clarify when the filtered wind direction is used and contrast it with the "current" yaw position (see revised paragraph in response to the previous comment).
* * *
RC5 page 7 line 25: The acronym SOWFA is quite well known in the community and should be mentioned here.
* * *
We have changed the sentence from: "… based on data from LES using NREL's Simulator fOr Wind Farm Applications tool…" to: "… based on data from LES using NREL's SOWFA (Simulator fOr Wind Farm Applications) tool…"
* * *
RC6 page 11 line 4f: To avoid confusion please mention, that ndelta is the Kronecker-Delta.
* * *
Good catch. We have added "…where $\delta(\cdot)$ is the Dirac delta function and…" after Eq. 5. We are calling this the Dirac delta function because the equations are written based on continuous time. Later we specify that the PDFs are discretized using a step size of 1 degree.
* * *
RC7 page12 line 1f: Which wind direction signal was used in the joint distribution in Fig. 7. The low-filtered wind direction or the "combined" wind direction?
* * *
Good question, and something we should have clearly stated. Starting in Fig. 7, the wind direction being plotted in all figures is the low frequency wind direction, which we expect to be more relevant as an input to FLORIS.

We now clarify that Fig. 7 is showing the distribution as a function of low frequency wind direction when introducing the figure on page 12: "…shown in Fig. 7, which compares theoretical joint PDFs of *low-frequency* wind direction and yaw position using Equations 4 through 7 with a histogram determined from simulation."

Additionally, this is clarified in the caption of Fig. 7: "Distributions of *low-frequency* wind direction and yaw position with yaw offset control…"

We now state in the rest of the Section 3 and Section 4 as well as in the captions for Figs. 8-13 that results are shown as a function of low-frequency wind direction. Furthermore, the math in Sections 3.1 and 3.2 now includes the correct low-frequency wind direction symbol: $\phi_l$

RC8 page 13 Equation (9) For easier readability in the equations, I would advise to only italicize variables, as the ISO standard suggests. The l (in nhatnphi_l), FLORIS and the d of the integrator should be written in roman.

Thank you for this information. We have now applied the regular Roman text style to the appropriate places in Eq. 9 as well as the rest of the math in the manuscript.

RC9 page 13 Section 3.3: Here the parameters for the uncertainty in the wind direction (x-axis Fig.8) and the yaw (y-axis Fig. 8) are tuned using the simulation. But in the simulation, the yaw uncertainty should either not exist or be adjustable. Can you explain the result for the yaw uncertainty? Is it possible that the hysteresis of the Yaw controller and the Yaw process used in the simulation influences the parameterization?

You are correct that yaw uncertainty, in the sense of accurately knowing the true yaw position, does not exist, or can be controlled, in the simulations. Therefore, yes, the yaw "uncertainty" we are modeling is due to the yaw controller dynamics and hysteresis, as you suggest. To clarify this point, we have added a discussion of the source of yaw uncertainty to the first paragraph of Section 3.3:

"Note that we assume the controller has perfect knowledge of the yaw position. Yaw uncertainty, as defined here, instead stems from the dynamics and hysteresis of the yaw and wake steering controllers. For example, the target yaw misalignment determined by the yaw offset controller lags behind the true wind direction, which can cause the yaw controller to settle on an unintended yaw position. Additionally, while the turbine is yawing to achieve a particular yaw misalignment, the offset target from the yaw offset controller can change, again causing the controller to stop at an unintended yaw position."

RC10 page 16 Figure 10: The normalized power shown here is probably the sum of both turbines and not just the turbine downstream. This should be clearly stated.

Yes, the power shown in all figures starting with Fig. 10 is the combined power of the two turbines. We now clarify this in the first sentence of Section 4.1:

[revised manuscript text omitted]